# Microscopic geared metamachines

Gan Wang[1], Marcel Rey [1,2], Antonio Ciarlo [1], Mahdi Shanei [3], Kunli Xiong [1,4], Giuseppe Pesce [1,5,6], Mikael Käll [3] & Giovanni Volpe [1] ✉

The miniaturization of mechanical machines is critical for advancing nanotechnology and reducing device footprints. Traditional efforts to downsize gears and micromotors have faced limitations at around 0.1 mm for over thirty years due to the complexities of constructing drives and coupling systems at such scales. Here, we present an alternative approach utilizing optical metasurfaces to locally drive microscopic machines, which can then be fabricated using standard lithography techniques and seamlessly integrated on the chip, achieving sizes down to tens of micrometers with movements precise to the sub-micrometer scale. As a proof of principle, we demonstrate the construction of microscopic gear trains powered by a single driving gear with a metasurface activated by a plane light wave. Additionally, we develop a versatile pinion and rack micromachine capable of transducing rotational motion, performing periodic motion, and controlling microscopic mirrors for light deflection. Our on-chip fabrication process allows for straightforward parallelization and integration. Using light as a widely available and easily controllable energy source, these miniaturized metamachines offer precise control and movement, unlocking new possibilities for micro- and nanoscale systems.

Geared mechanisms − systems where interconnected gears transfer work and perform mechanical tasks − have long mirrored the advancement of human technology. Their evolution spans from the wind and water mills of ancient times to the steam engines of the Industrial Revolution and on to modern automotive, aerospace, and robotics applications[1]. Current advances focus on miniaturizing these gears to micrometer scales[2]. While enhancing material efficiency and reducing waste, this miniaturization also opens new possibilities for mechanizing and exploring a length scale that has largely remained elusive. For example, down-sizing geared mechanisms provides tools to gain a deeper understanding of microscopic phenomena such as friction[3,4] and surface interactions[5,6], while driving technological innovations such as high-performance microfluidic devices[7–13] and reconfigurable optical technologies[14–16]. Moreover, innovations in both manufacturing and powering these geared systems are also impacting fields such as microrobotics[17,18], optical systems[19], and force sensors[20].

Efforts to miniaturize gears have primarily focused on creating individual micromotors − microscopic objects capable of rotation. Various mechanisms have been explored to power these devices, including static[21–24] and AC electric fields[25–28], magnetic fields[29–33], and light fields[34–37]. However, incorporating these micromotors into functional microscopic geared mechanisms remains a significant challenge. Traditional semiconductor manufacturing methods for electrostatically driven gears[38,39] are hindered by the need for electric connectors, which occupy considerable space around each micromotor, limiting both miniaturization and parallelization. While far-field approaches such as AC electric, magnetic, and light fields allow further miniaturization of individual micromotors (Table S1), they present their own limitations in achieving the integration of multiple motors into on-chip complex geared mechanical machines, which can be defined as at least two mechanical parts interacting with each other to form a cohesive unit capable of generating and trasmitting work.

[1]Department of Physics, University of Gothenburg, Gothenburg, Sweden. [2]Institute of Physical Chemistry, University of Münster, Münster, Germany. [3]Department of Physics, Chalmers University of Technology, Gothenburg, Sweden. [4]Department of Materials Science and Engineering, Solid State Physics division, Uppsala University, Uppsala, Sweden. [5]Department of Physics E. Pancini, University of Naples Federico II, Complesso Universitario Monte Sant'Angelo, Naples, Italy. [6]Department of Neuroscience, Reproductive Sciences and Dentistry, University of Naples Federico II, Naples, Italy. ✉e-mail: giovanni.volpe@physics.gu.se

Methods employing AC electric[25–28] or magenetic fields[29–33] require field-responsive materials (e.g., conductive or magnetic components), complicating integration into mechanical machines and making systems prone to external interference. Integrating multiple components further risks cross-interference, potentially degrading functionality; moreover, these approaches often lack the ability to address gears individually[40]. Light-based methods[9,10,41], like optical tweezers, are less constrained by material properties but require focused light beams, limiting their large-scale manipulation potential. While methods based on photo-generated electric fields[13] and light-driven chemical reactions[42,43] offer promising solutions, they typically lack flexibility and are restricted to specific chemical environments. As a result, a scalable approach for microscopic geared mechanisms that overcomes all these limitations remains elusive (Supplementary Table 1 and Supplementary Note: Comparison of Mechanisms to Drive Micromotors).

Recent advances in active matter have used unfocused light to propel microscopic vehicles employing plasmonic or dielectric metasurfaces that generate lateral optical forces through directional light scattering. For example, it has been shown that microvehicles with plasmonic or dielectric nanostructures arranged in a parallel pattern can move forward under linearly polarized light via linear momentum transfer[44,45] and can be steered using polarized light through spin angular momentum transfer[45]. Furthermore, it has also been shown that arranging the scatterers in a circular pattern enables rotation under linearly polarized light[44,46]. More advanced designs incorporate four individually addressable chiral plasmonic nanoantennas, allowing full 2D motion control through the application of dual-wavelength light[47].

Here, we build on these recent advances to fabricate geared mechanism driven by optical metasurfaces that operate under uniform illumination. Using silicon as the primary material ensures compatibility with standard photolithography, facilitating large-scale manufacturing. This approach creates a versatile platform for precise control and movement of geared functional devices, enabling unprecedented capabilities in micro- and nanoscale mechanical systems.

## Results

### Micromotors powered by optical metasurface

We present a rotating micromotor powered by an optical metasurface. This micromotor is constituted by a metarotor − a ring structure containing a metasurface − that is securely anchored to a glass chip using a capped pillar, as shown in Fig. 1a.

The fabrication process involves four key steps. First (Fig. 1b), the metasurface is etched, optimized for operation in water under 1064 nm plane wave illumination. The metasurface's unit cells, or meta-atoms, are composed of two asymmetric rectangular Si blocks, with dimensions 270 nm × 200 nm × 460 nm and 400 nm × 200 nm × 460 nm, respectively, separated by a subwavelength gap of 50 nm to maximize the efficiency of the +1 light diffraction order relative to the 0 and −1 orders, based on the design presented in ref. 45 (see more details in Supplementary note "Design principle of metasurface"). Next, a $SiO_2$ ring is etched to support the metasurface (Fig. 1c). This is followed by the fabrication of the SU-8 pillar (Fig. 1d) and its cap (Fig. 1e). The capped pillar anchors the metarotor to the $SiO_2$ substrate, allowing it to rotate freely while suspended in water. The final step involves removing a sacrificial layer between the ring and the substrate. Using this approach, tens of thousands of micromotors can be fabricated within a 5 mm × 5 mm area on a single chip. Moreover, the fabrication process can be scaled up to the wafer level. A comprehensive explanation is provided in the Methods section and Supplementary Figs. 1 and 2.

This metarotor is powered by the interaction between the incoming light and the metasurface. By deflecting the incoming light, the metasurface induces a force on the rotating ring in the opposite direction due to the conservation of linear optical momentum, as schematically shown in Fig. 1f. By tuning the design of the metasurface, we can control the details of this interaction and therefore the metarotor rotation.

The metasurface of the metarotor in Fig. 1a–e comprises four segments, each with meta-atoms arranged in parallel but rotated by 90° relative to the adjacent segments. The direction of the resulting forces($F$) acting on the metarotor under uniform linearly polarized light for each segment are schematically depicted using white arrows in Fig. 1g. The forces ($F$) have a non-zero lever arm ($r$) relative to the common center of mass, generating a torque $\tau = rF$ that induces a counterclockwise rotation of the metarotor, as depicted by the black arrow in Fig. 1g. (more details in Supplementary note " Rotation mechanism of the motor" The resulting movement is illustrated in Fig. 1h and Supplementary Video 1.

We can control the metarotor's angular velocity ($\omega$) either by altering the metasurface design or by modifying the light intensity. Figure 1i compares metarotors of equal dimensions but varying numbers of meta-atoms under different light intensities. The angular velocity increases with more meta-atoms (Supplementary Fig. 3 and Supplementary Video 2). At low intensities (12.7 µW µm$^{-2}$, 30.3 µW µm$^{-2}$, 48.0 µW µm$^{-2}$), this increase is linear. However, at higher intensities (70.8 µW µm$^{-2}$, 88.5 µW µm$^{-2}$), the relationship becomes nonlinear due to increased light absorption by the meta-atoms, leading to a local rise in water temperature, reduced viscosity, and decreased rotational viscous drag ($\gamma_r$) (Supplementary Fig. 4). At low intensities, momentum transfer is the primary factor, while at high intensities, both decreased viscous drag and increased momentum transfer create nonlinear effects (Supplementary Fig. 5). Consequently, a nonlinear relationship between the angular velocity and light intensities of different meta-atoms is also observed in Fig. 1j (Supplementary Figs. 6–8 and Supplementary Video 3). Under the intensity of 88.5 µW µm$^{-2}$, we measure a maximum torque of 36 pN · µm obtained from on $\tau = \omega\gamma_r$. It is important to note that there is an upper limit to the applicable torque, as higher laser powers can induce local heating and bubble formation (see Supplementary Note "Rotation mechanism of the motor" for details). The energy conversion efficiency ($\epsilon$) of the motor can then be calculated as $\epsilon = P_{out}/P_{in} = \tau\omega/P_{in}$. The order of magnitude of the conversion efficiency is approximately 10$^{-14}$, which is consistent with previously reported motors driven by light momentum[46,47]. A comparison of efficiencies is provided in Table S1. Regarding motor durability, although the chip is not yet optimally packaged for long-term stability, the motor remains operational under continuous illumination for up to eleven hours (Supplementary Video 4). Furthermore, it does not undergo structural degradation even when irradiated for eleven hours and stored for up to six months (Supplementary Fig. 9). Nevertheless, while the motor is in operation, its rotational speed gradually decreases and eventually the motor stops. This is likely due to changes in the solution environment (such as local surfactant redistribution and the accumulation of impurities) leading to increased friction at the motor-substrate interface. Notably, the motor can resume rotation after gentle cleaning and solution exchange, indicating that these effects are reversible and can be mitigated through improved packaging and fluid handling.

Finally, we explore the influence of the gap size between the metarotor and the capped pillar, as shown in Fig. 1k (Supplementary Fig. 10 and in Supplementary Video 5). A minor reduction in angular velocity with an increasing gap size is identified, remaining within the margin of error. Interestingly, the metarotors experience enhanced confinement with a reduced gap size, evident from the probability distribution of the metarotors relative to the center of the pillar (Fig. 1l).

Metarotors can be fabricated in various sizes. We have built a metarotors down to a smallest size of 8 µm in diameter, as shown in Supplementary Video 6. Multiple metarotors rotate stably and simultaneously under uniform illumination (Supplementary Video 7).

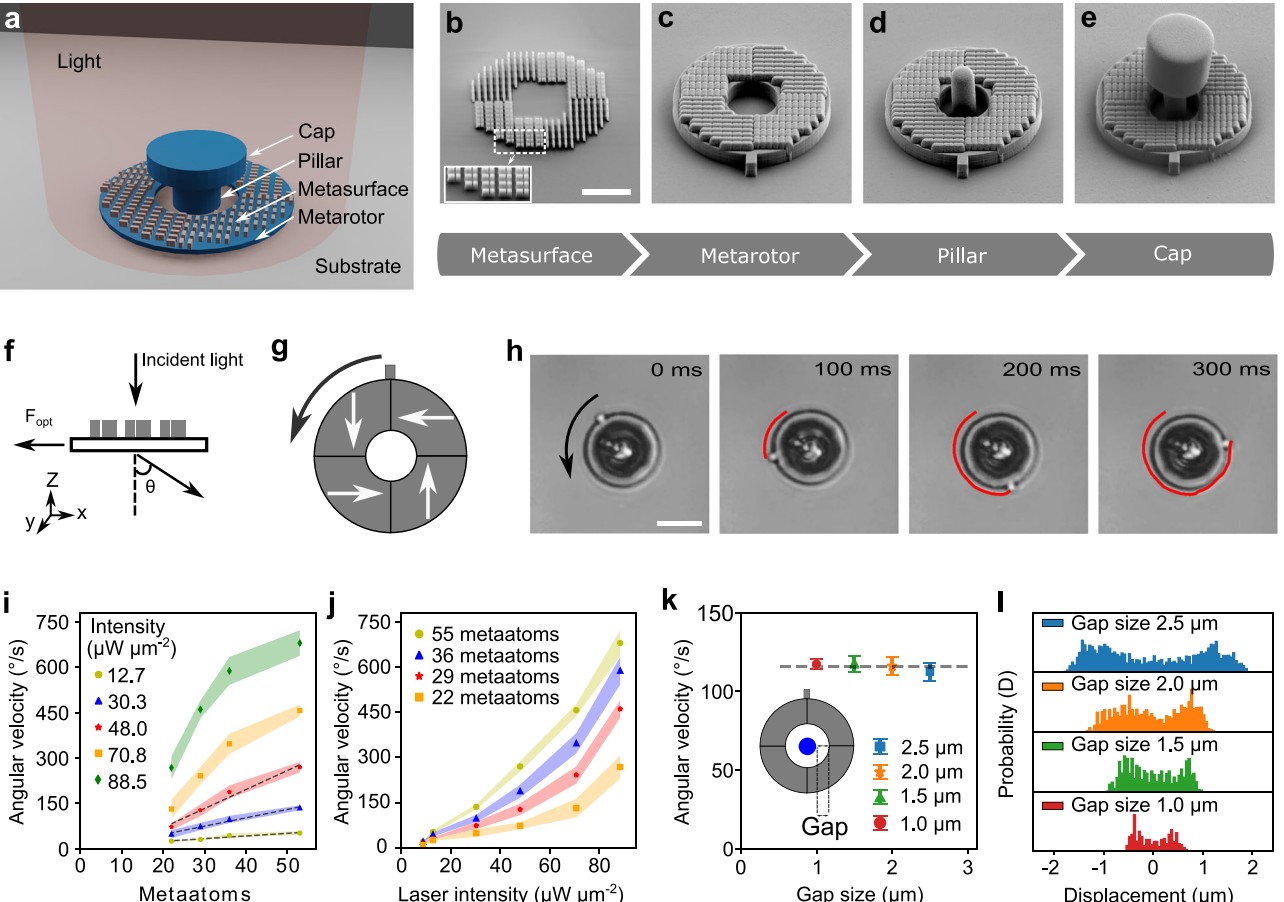

**Fig. 1 | Metarotors. a** Schematic illustration of the optically powered micromotor, featuring a ring-shaped metarotor containing a metasurface, anchored to a glass chip using a capped pillar. **b**–**e** Scanning electron microscopy (SEM) images documenting the micromotor fabrication process (scale bar: 5 μm): **b** A Si metasurface is nanofabricated via electron beam lithography; the inset zooms in on the meta-atoms constituting the metasurface. **c** A SiO₂ ring containing the metasurface is etched. **d** The central pillar and **e** the cap are fabricated with SU-8 microlithography. **f** Illustration of the deflection of light by the metasurface, resulting in a force acting on the metasurface in the opposite direction. **g** The metasurface consists of four segments with different orientations of the meta-atoms. The white arrows indicate the forces they exert onto the metarotor upon illumination with linearly polarized light. The black arrow represents the resulting counterclockwise rotation. **h** Optical microscopy images (see Supplementary Video 1) of the rotation of a metarotor under a linearly polarized light beam with an intensity of 35 μW μm⁻². The red line indicates the tracking of a protrusion on the metarotor outer border. Scale bar: 10 μm. Average angular velocities of metarotors with equal diameters (16 μm) illuminated by a linearly polarized plane light beam as a function of the number of **i** meta-atoms and **j** laser light intensity. The shaded regions indicate the standard deviation. **k** Independence of the angular velocity from the gap size between the ring and the pillar. Error bars represent standard deviation from three measurements for each condition. **l** Probability distributions of the metarotor position along the x-axis for varying gap sizes. Smaller gap sizes lead to a higher confinement.

Increasing the spot size and/or laser power enables control of more micromotors over a larger area.

## Gear trains powered by metarotors

Our next step is to employ metarotors to actuate microscopic machines, transferring their work through gear trains. In the first application, a metarotor serves as a driving gear with a ring diameter $D_m$ (a metagear, Fig. 2a) to propel a driven gear with a ring diameter $D_p$ (Fig. 2b–d, Supplementary Video 8). The metagear is fabricated similar to the metarotor described in Fig. 1b–e, with the addition of teeth to convert the ring into a gear. The driven gear is also fabricated like the metarotor but with added teeth and without the metasurface.

As expected, the angular velocity of the passive driven gear, $\omega_p$, depends on the angular velocity of the motor metagear, $\omega_m$, according to the ratio of their diameters:

$$\omega_p = \frac{D_m}{D_p} \omega_m. \tag{1}$$

This theoretical relationship is depicted by the dashed line in Fig. 2e, with the experimental data represented by their corresponding symbols. This system demonstrates the ability to multiply torque when $D_p > D_m$ or speed when $D_p < D_m$.

The metagear can also be used to actuate multiple driven gears at once, as shown in Fig. 2f–h and Supplementary Video 8. In these examples, all gears have the same diameter and therefore the same angular velocity $\omega = \omega_m = \omega_p$. With the addition of each extra driven gear, $\omega$ decreases. We expect this decrease to depend inversely on the number of gears $N$ (i.e., $\omega \propto \frac{1}{N}$, where $N$ is the number of gears) due to increased friction, as shown by the dashed line in Fig. 2i. In fact, the experimental results, shown by the symbols in Fig. 2i, indicate that $\omega$ decreases faster than expected as $N$ increases, probably because each additional gear introduces new contact surfaces that contribute to higher frictional forces within the system.

The use of gear trains on the microscale offers the design flexibility needed to create more complex machines. Examples of gear train configurations with different gear geometries are showcased in Supplementary Video 9.

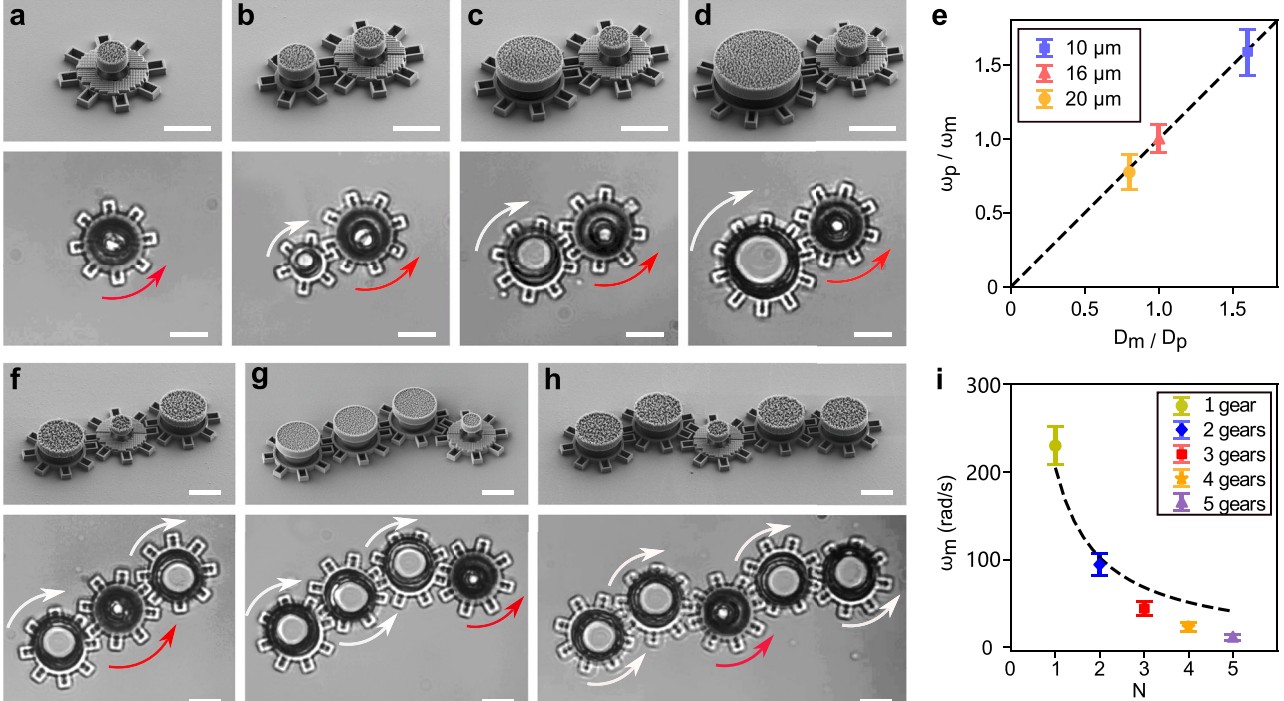

**Fig. 2 | Gear trains powered by metarotors. a–d** SEM images (top panels) and optical microscopy images (bottom panels) of metarotors acting as driving gears that propel passive gears with varying diameters (**b**–**d**). The rotation of the metarotors is indicated by the red arrows, while the rotation of the driven gear is indicated by the white arrows. **e** The average ratio of angular velocity between the driving gear ($\omega_m$) and the driven gear ($\omega_p$) depends on the ratio of their diameters. **f**–**h** SEM images (top panels) and optical microscopy images (bottom panels) of a single driving gear actuating a train of driven gears with the same diameter: **f** $N = 3$, **g** $N = 4$, and **h** $N = 5$ total gears (including the driving gear) with the same diameter $D_m = D_p$. **i** The angular velocity of the driven gears $\omega = \omega_m = \omega_p$ versus the number $N$ of gears in the gear train. The dashed line represents $\omega/N$. The optical microscopy images are from Supplementary Video 6. Error bars in **e** and **i** represent standard deviation from three measurements for each condition. Scale bars: 10 μm.

## Controlling the metarotor rotation velocity and direction

Until now, we have used only linearly polarized light to power the metarotors. Remarkably, by altering the polarization of the illumination, we can also dynamically control their rotational direction.

We first examine a metarotor with metasurface segments generating clockwise forces (Fig. 3a–d). The metarotor ring is divided into eight segments with only four containing metasurfaces, as shown by the shaded areas in the schematics on the top of Fig. 3a–c; the white arrows indicate the forces generated by each segment under the various illumination conditions, while the black arrow represents the overall metarotor rotation direction (which is always clockwise for this metasurface design) and magnitude (which varies for different illumination polarizations). A measured 5-second trajectory is overlaid on the optical images below each schematic in Fig. 3a–c. Under linearly polarized light, this metarotor rotates clockwise (Fig. 3a). A right-hand circular polarization increases the metarotor angular velocity (Fig. 3b), while a left-hand circular polarization decreases it (Fig. 3c). This variation is due to the superposition of the change in momentum as the incident light is deflected by the metasurface and of spin angular momentum (SAM) transferred to the metarotor[48,49].

Next, we consider a specularly symmetric design of the metasurfaces, as shown in the schematics in Fig. 3e–g. This design results in opposite forces and therefore a counterclockwise rotation. For linear polarization (Fig. 3e), the metarotor exhibits a rotational velocity similar to that in the previous case (Fig. 3a) but in the opposite direction. Under circular polarization (Fig. 3f, g), the rotational velocity decreases for right-handed polarization and increases for left-handed polarization. This is the opposite behavior to that observed in the previous case (Fig. 3b,c), which can also be observed when varying the illumination polarization (Fig. 3h). This is due to the fact that, while the

metasurface light deflection is the opposite than in the previous case, the transfer of SAM to the metarotor is the same as in the previous case for each polarization.

In order to build a metarotor that can rotate in either direction depending on polarization, we can now combine these two designs as shown in Fig. 3i–k. Now, under linear polarization, the forces cancel out, resulting in no movement (Fig. 3i). Under right-hand circular polarization, the metarotor rotates clockwise (Fig. 3j), and, under left-hand circular polarization, it rotates counterclockwise (Fig. 3k). This permits to dynamically adjust the rotation direction and velocity of the metarotor by the illumination polarization (Fig. 3l).

Conveniently, these designs can be fabricated on the same chip, enabling distinct behaviors by changing the light polarization (Supplementary Video 10). This capability extends to altering the rotational direction within gear trains as well (Supplementary Video 11).

## Responsive metamachines transferring rotational to linear movement

Having developed metarotors capable of dynamically altering their rotation direction, we use them to build some microscopic machines with movable parts also to convert rotational motion into linear motion in a rack and pinion design (Fig. 4). In this design (Fig. 4a–c), the metarotor acts as the pinion and engages a rack. When exposed to right-handed circularly polarized light, the metarotor rotates clockwise, moving the rack to the right. Conversely, under left-handed circularly polarized light, the metarotor rotation direction is reversed, causing the rack to move to the left. By alternating these two polarizaitons, we can obtain an oscillatory back-and-forth motion of the rack, as shown in Fig. 4d (and Supplementary Video 12). The amplitude and frequency of this motion can be

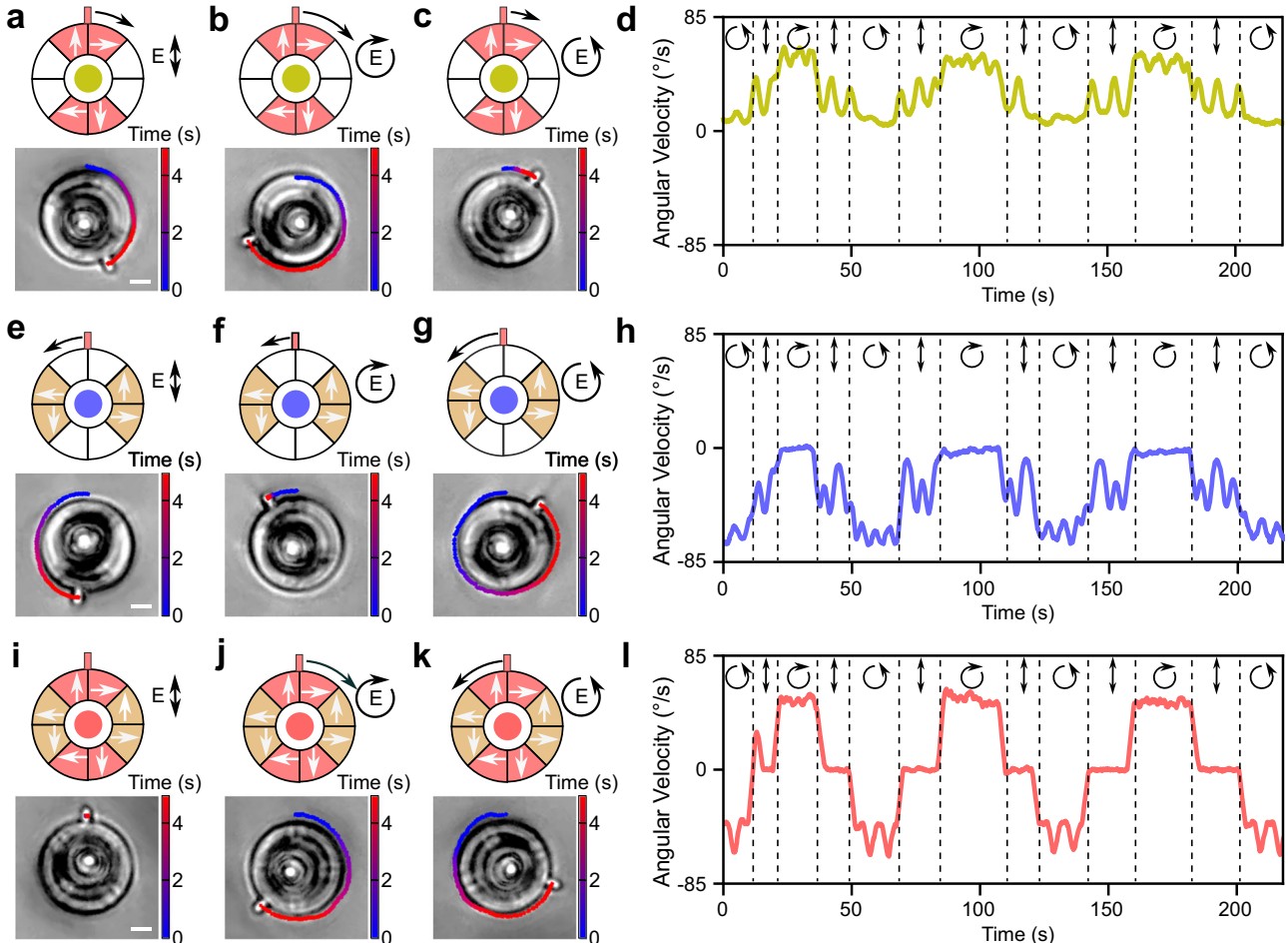

**Fig. 3 | Metarotor control by metasurface design and light polarization.**
**a**–**d** Metarotor with a metasurface design that generates clockwise rotation as a function of light polarization. Schematic illustration of the metasurface design (top) and optical microscopy images (bottom) under illumination with **a** linear polarization, **b** right-hand circular polarization, and **c** left-hand circular polarization. The metasurface segments are indicated by the red-shaded areas and the resulting forces by the corresponding white arrows. The global rotation direction and speed of each motor is indicated by the direction and length of the black arrows. The experimentally measured rotation of the metarotor is overlaid on the optical images. **d** Angular velocity of the motor with changing light polarization. **e**–**h** Metarotor with a metasurface design that generates counterclockwise rotation as a function of the light polarization. **i**, **j** Metarotor with a metasurface design combining the designs in (**a**–**c**) and (**e**–**g**) as a function of the light polarization. The metarotor remains stationary under linear polarization, rotates counterclockwise under left-hand circular polarization, and clockwise under right-hand circular polarization. The optical microscopy images are taken from Supplementary Video 8. Scale bars: 5 μm.

adjusted by controlling the light intensity and the intervals between changes in polarization.

Next, we can consider an alternative design (Fig. 4e–g) that will permit us to oscillate the rack under constant linearly polarized light. To achieve this, we add a metasurface to the rack, as shown by the red-shaded area in Fig. 4e. The metarotor, now featuring a single tooth, moves the rack leftward when engaged, while the rack metasurface pushes it rightward, mimicking the behavior of a microscopic spring. A delicate balance between these forces, which can be optimized by the number of meta-atoms (Supplementary Fig. 11, Supplementary Video 13, Supplementary Video 14), results in an oscillatory back-and-forth movement of the rack under constant linearly polarized light (Fig. 4h). While this design offers comparatively lower tunability and flexibility compared to the previous one, its advantage lies in obtaining an oscillatory rack motion without the need to repeatedly alter the polarization of the light.

Finally, we demonstrate how this design can be used to control the movement of some microscopic gold mirrors and therefore the reflection and transmission of light on the microscopic scale (Fig. 4i–l). By embedding gold mirrors in the rack, indicated by the golden yellow

areas in the schematic in Fig. 4i, we can dynamically block and deflect light within an optical device (Fig. 4l, Supplementary Video 15). Our system enables two-dimensional mirror shifts via light, achieving displacements up to tens of micrometers. It can serve as an alternative to established electrostatic actuator micromirrors for applications requiring larger displacement, expanding possibilities for dynamic or fluidic applications.

In comparison with previously reported field-driven micromotors[25–37], this highlights the advantage of adopting a design and fabrication process compatible with metal-oxide-semiconductor (CMOS) technology, as it permits us to ensure a straightforward implementation with adherence to reproduction standards in cleanroom facilities as well as seamless parallelization of micromotors on a single chip. Additionally, it enables the direct integration of various other CMOS-based components, further amplifying the versatility and efficiency of our advanced manufacturing approach.

## Discussion
In summary, our work introduces optically powered micromotors capable of performing work under broad light illumination, utilizing

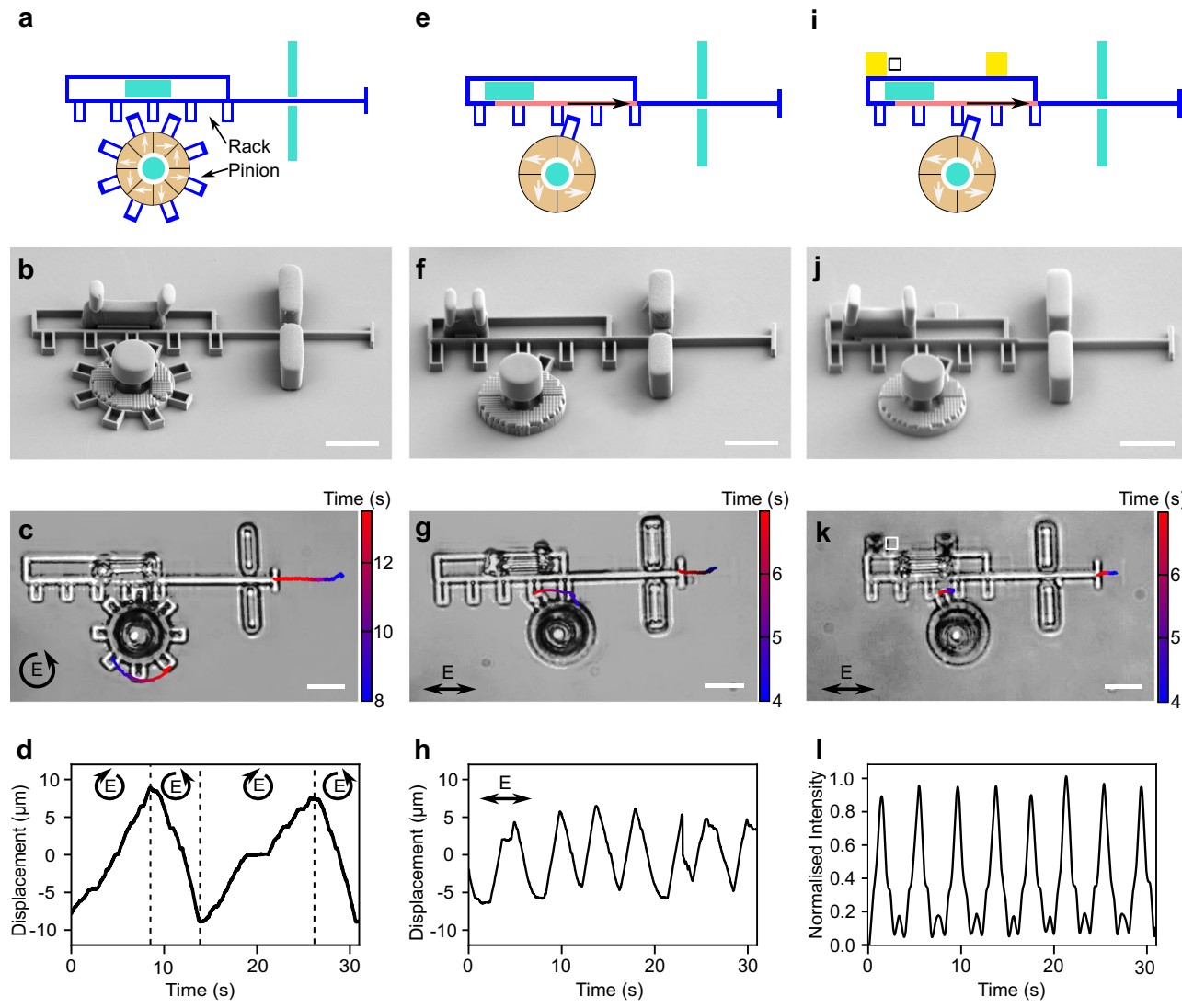

**Fig. 4 | Microscopic rack and pinion metamachines. a, e, i** Schematic illustrations of three designs of rack and pinion metamachines to convert the rotational motion produced by a metarotor into linear motion. The movable rack and pinion are shown in blue, while the immobile parts are in turquoise. Metasurface segments are highlighted in red and yellow. The forces generated on the driving gear and rack under illumination are depicted by white and black arrows, respectively. **b, f, j** Corresponding SEM images, and **c, g, k** optical microscopy images from Supplementary Videos 10, 12, 13. **d** The pinion metarotor is equipped with a metasurface designed so that its rotation direction is different for left and right-handed circularly polarized light, enabling forward motion of the rack under right-handed circular polarization and backward motion under left-handed circular polarization, allowing dynamic back-and-forth motion by changing the circular polarization of light. **h** Equipping both the rack and the pinion with metasurfaces permits oscillatory rack movement under constant linearly polarized light. The pinion has a single tooth that periodically moves the rack leftward when engaged, while the rack's metasurface moves it rightward, mimicking a macroscopic spring. Balancing the forces of both metasurfaces achieves oscillatory rack movement under linearly polarized light. **l** The same rack and pinion design can periodically move a gold mirror (illustrated in golden yellow in **i**), changing the average amount of transmitted light **l** through the white box shown in **i** and **k**. Scale bars: 10 μm.

integrated metasurfaces. This innovation achieves motor miniaturization to diameters below 10 μm, enabling straightforward parallelization and precise control over individual micromotor units. Control over rotation direction and speed can be achieved through metasurface design or adjustments in light intensity and polarization.

Furthermore, these micromotors can be assembled into functional microscopic metamachines, transducing rotational motion in gear trains or converting it to linear motion using rack and pinion designs. The on-chip fabrication process, compatible with standard CMOS lithography, facilitates seamless integration with other CMOS components like metalenses and plasmonic sensors. Future developments may involve arrays of micromotors for collective manipulation of objects and flow control on the micron scale.

The use of phase-transition materials[50] (e.g., VO₂) could be integrated into the metasurface design, enabling real-time reconfiguration of optical properties in response to external stimuli such as temperature, electric fields, or light[51]. This would address the current limitation of relying on pre-designed metasurfaces, which restricts dynamic motion adjustability. Additionally, adaptive optics, including deformable mirrors or spatial light modulators, could enhance flexibility by enabling precise wavefront correction and dynamic light modulation.

Beyond this, incorporating involute or cycloidal tooth profiles, commonly used in macroscopic gears, could reduce inter-gear friction and improve transmission efficiency. Alternative metasurface materials, such as TiO₂, could extend the operational wavelength into the visible light region, simplifying optical calibration. These

advancements could improve the system's performance, adaptability, and applicability across diverse environments.

By using light as a widely available and biocompatible energy source, these micromo- tors are well-suited for manipulating biological matter, including bacteria and cells. The system employs a 1064 nm laser, which minimizes damage to biological samples due to its low absorption by water and tissues[45,52]. The light can be focused from a large area onto the small driving gear, operating at a low power requirement of just a few mW, which remains well within the safe thresholds for biological systems. Importantly, the light can be selectively directed to the driving gear, allowing it to mechanically actuate passive structures without directly exposing biological samples to the light source. This non-toxic, indirect energy delivery mechanism broadens the applications of our light-driven micromotors and metamachines in biomedical environments.

Looking ahead, we anticipate integrating our metamachines with planar optical elements, such as high-numerical-aperture metalenses. These metalenses can focus incoming light, enabling precise manipulation of colloidal objects. This synergy could lead to advanced applications, such as mechanically programmable devices that alter optical properties through planar optical elements, including the generation of spatially structured lights. Our metasurfaces can elevate optical force modulation to the femtonewton (fN) scale, making them valuable on-chip force measuring instruments for assessing mechanical properties of individual cells or biological macromolecules, such as DNA.

## Methods

### Metarotor fabrication protocol
The fabrication of optically powered microscopic motors and micromachines relies on cleanroom fabrication methods, which can be divided into four steps: preparation of the substrate, fabrication of the metasurface, etching of the metarotor bearing the metasurface, and fabrication of the pillar and of the cap to anchor the ring. The details are schematically illustrated in Supplementary Fig. 1.

**Preparation of the substrate.** For the fabrication, we used a 4-inch fused silica wafer. An 800 nm layer of amorphous silicon (a-Si) was deposited using low-pressure chemical vapor deposition (LPCVD, Furnace Centrotherm) to serve as a sacrificial layer, facilitating the later release of the ring from the substrate. Subsequently, a 400 nm layer of $SiO_2$ was deposited via plasma-enhanced chemical vapor deposition (PECVD, Oxford PlasmaPro 100), serving as the material for crafting the metarotor. Finally, a 460 nm a-Si layer was deposited using LPCVD (Furnace Centrotherm), to be etched to fabricate the metasurface.

**Fabrication of the metasurface.** To pattern the a-Si for metasurface fabrication, a 300 nm thick layer of ARP6200 (CSAR ARP 6200.13) resist was spin-coated on the substrate. Subsequently, electron beam lithography (Raith EBPG 5200) was employed for exposure and the resist was developed using a developer (n-Amyl acetate). Next, a metal layer composed of 2 nm chromium (Cr) and 40 nm nickel (Ni) was evaporated (Kurt J. Lesker PVD225) onto the patterned resist. This was followed by the subsequent lift-off process in an organic remover (Microposit Remover 1165). The metal layer served as a hard mask for undergoing the chlorine ($Cl_2$) reactive ion etching process (Oxford Plasmalab 100), enabling the precise etching of a 460 nm a-Si metasurface. After completing this process, the metallic layer was removed using a chemical etching method (SunChem Nickel/Chromium etchant). Finally, a 550 nm $SiO_2$ layer was deposited onto the metasurface using PECVD (Oxford plasmaPro 100) to serve as a protection layer.

**Fabrication of the metarotor and movable parts.** For the metarotor and the movable parts of the geared micromachines, two patterning methods were used depending on the required precision. For high-precision parts, EBL (Raith EBPG 5200) was used to define the pattern on the resist ARP6200 (CSAR ARP 6200.13). A metal layer consisting of 2 nm Cr and 60 nm Ni was evaporated (Kurt J. Lesker PVD225) onto a patterned resist, followed by the subsequent lift-off process in Remover-1165 (Microposit Remover 1165). For parts tolerating lower precision, direct laser writing (Heidelberg, MLA 150) was employed to define a double-layer photoresist, LOR3A/S1805. LOR3A (MicroChem Photoresist) acted as a sacrificial release layer for the lift-off process, while S1805 (Shipley Photoresist) served as the top positive resist. The exposed resist was developed using a developer (Microposit MF-CD 26) and then employed for lifting off the deposited 2 nm Cr and 60 nm Ni as a durable hard mask for the subsequent etching step. In both cases, a reactive ion etching process (Oxford Plasmalab 100) with a mixture of fluoroform ($CHF_3$) and argon (Ar) gases was employed to etch the $SiO_2$ layer. Following this, $Cl_2$ reactive ion etching (Oxford Plasmalab 100) was applied to remove the exposed a-Si between the PECVD-deposited $SiO_2$ and the fused silica substrate.

**Fabrication of the pillar, cap, and immobile parts.** The immobile components, including the pillars and caps employed to anchor the micromotors and micromachine parts, were produced through a direct laser writing process. Initially, a 3.8 μm layer of SU-8-3005 (MicroChem Photoresist), a negative photoresist, was spin-coated onto the substrate. Direct laser writing (Heidelberg, MLA 150) was then utilized to expose the photoresist, defining pillars in the center of the rotating rings and micromachine parts. After developing the SU-8 3005 in Remover500 (Microresist Technology), the sample underwent a 10-minute hard bake at 160 °C to enhance the durability of the pillars.

Subsequently, a 4 μm layer of positive photoresist AZ4533 (Clariant Photoresist) was spin-coated onto the substrate as a sacrificial layer to create caps on the pillars. A 250 W $O_2$ plasma (Plasma-Therm) was applied for 1 minute to ensure that the upper surface of the SU-8 pillar remained uncovered by any AZ4533 residues, ensuring successful linking of the pillar and cap. The cap was fabricated by spin-coating a 3.8 μm SU-8-3005 layer on top of the sacrificial AZ4533 layer, which was then defined using direct laser writing. SU-8 3005 was initially developed in Remover500, after which AZ4533 was removed by immersing the sample in acetone (Sigma Aldrich) to separate the cap from the disks. In the final step, the rotating rings and micromachine parts were released from the substrate by removing the sacrificial a-Si layer using highly selective xenon(II) fluoride ($XeF_2$) ion etching.

**Fabrication of gold mirrors.** The fabrication of the gold mirrors took place before depositing $SiO_2$ on the metasurface. The structure of the mirrors was defined using direct laser writing (Heidelberg, MLA 150) on the substrate coated with a double-layer of photoresist LOR3A/S1805, achieved through spin-coating. The exposed resist was developed using a developer (MF-CD 26), followed by the deposition of a 60 nm gold layer onto the resist (Kurt J. Lesker PVD225). Subsequently, the gold mirrors were formed by conducting a lift-off process to remove the unexposed photoresist in Remover 1165. The subsequent fabrication processes were then carried out as previously described.

**Further optimization of the fabrication process.** In this work, tens of thousands of micromotors were fabricated on a 5 mm × 5 mm area (Supplementary Fig. 2). The metasurface was created using electron beam lithography (EBL), while direct laser writing was used for the micromotor pillars and caps. These lab-based techniques offer design flexibility and rapid iteration but are highly time-consuming for wafer-scale manufacturing, requiring days to complete. Foundry-compatible alternatives, such as deep ultraviolet (DUV) lithography and nanoimprinting[53,54], can replace EBL for metasurface fabrication, while lithography and imprinting techniques can substitute direct laser writing for pillar and cap fabrication[55], enabling cost-effective scalability.

### Scanning electron microscopy imaging

For scanning electron microscopy (SEM) imaging, the samples underwent sputter-coating with a 15 nm gold layer. SEM images were captured using a Zeiss Supra 55, operating at a current of 5 kV, and an SE2 detector.

### Optical setup

A schematic of the optical setup is shown in Supplementary Fig. 10. The optical setup consists of two parts: the bright–filed home–built optical microscope to observe the motion of micromotors and micromachines; and the illumination part based on a 1064 nm laser to make the micromotors move. The incident laser beam is weakly focused from the upper surface of the chip using a lens, generating a beam with an approximate waist diameter of 300 μm on the chip. By using half-wave and quarter-wave plates, we controlled the incident light polarization, while the power of the laser on the sample is regulated manually using the combination of a half-wave plate and a polarizing beam splitter after the laser. The videos were acquired by using a CCD camera.

### Measurements

The measurements were conducted by creating a 4 μL sample cell, which consisted of the chip as bottom slide, a 120 μm thick poly-dimethylsiloxane (PDMS) spacer, and a coverslip. The cell was filled with an aqueous solution containing 0.005 wt% of Triton X-100 (Sigma Aldrich) to prevent the spinning disks from sticking to the substrate. Additionally, the chip was briefly inserted into an ultrasonic bath for 10 s to ensure none of the motors were stuck. The positions and orientations of microrotors or micromachines were examined through a custom-written Python code.

### Photothermal heating simulation

The temperature profile around the micromotor is evaluated by modeling the a-Si meta-atoms as the exclusive sources of photothermal heating, incorporating a complex refractive index $n = 3.8 + i0.0064$ as determine from ellipsometry. The heat source density $Q$ is derived through simulations of absorption cross-sections via field element calculations in COMSOL. The model include both heat conduction and convection, with room tempearture (293 K) set as the boundary condition. For a-Si, a thermal conductivity is 1.8 W/m · K and density is 2.329 kg m$^{-3}$.

## Data availability

The data generated in this study are provided in the Supplementary information/Source data file. Source data are provided with this paper.

## Code availability

The code used during this study is available from the corresponding authors upon reasonable request.

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

## Acknowledgements

The authors would like to acknowledge funding from the H2020 European Research Council (ERC) Starting Grant ComplexSwimmers (Grant no. 677511) (GV), the Horizon Europe ERC Consolidator Grant MAPEI (Grant no. 101001267) (GV), the Marie Sklodowska-Curie Individual Fellowship (Grant no.101064381) (MR), German Research Foundation (DFG)–SFB 1459/2 2025–433682494" (MR), and the Knut and Alice Wallenberg Foundation (Grant No. 2019.0079) (GV and MK). Fabrication in this work was done at Myfab Chalmers.

## Author contributions

G.W. and G.V. conceived the idea, G.W. developed the fabrication process, performed the experiments, and analyzed the data. M.R. and K.X. assisted with fabrication and measurements. G.P., A.C., and G.W. built the optical setup. M.S. conducted the thermal absorption simulation. G.V. and M.K. supervised the study. G.W., M.R., and G.V. wrote the manuscript, with contributions from all other co-authors.

## Funding

## Competing interests

The authors declare no competing interests.
