## [Transparent Peer Review file · Nature Communications]

Microscopic Geared Metamachines

Corresponding Author: Professor Giovanni Volpe

Version 0:

Reviewer comments:

Reviewer #1

(Remarks to the Author)

The work by Wang incorporated silicon metasurface to improve the light-matter interaction for light-driven micromotors. The authors have fabricated the meta-motor using photolithography and laser direct printing. They investigated the rotation velocity and direction at different laser intensities, geometries, and incident laser polarizations. They also demonstrated the gear trains powered by the metamotors, as well as the translation from rotational to linear motion. I have a couple of comments.

1. The use of metasurface in light-driven micromachines is significant, but not new. A couple of works have been published in the past several years, such as Nature Nanotechnology volume 17, pages477–484 (2022), Nature Nanotechnology volume 16, pages970–974 (2021), etc. I noted that the authors have cited the second paper. I did not see the superiority of the designed metasurface/metamotors over those in these published works, regarding the functionalities and the efficiency of the micromachines. The authors should mention these relevant works and better place their work in the context.
2. The key to incorporate metasurface in light-driven micromachines is to introduce additional functionalities beyond those without metasurface, or to improve the optomechanic coupling efficiency of the devices. However, I did not see any discussion in both the main text and the SI on the analysis of light-metasurface interaction and generation of optical force. What is the design principle of the metasurface? Is it the optimized design? What is the optomechanic coupling efficiency? Is silicon the optimized materials? How about plasmonic metasurface which could improve the local electric field? These questions should be carefully addressed as they seem to me much more important than the control experiments given in the main text.
3. The author claimed in the introduction along the line of miniaturization of micromotors. However, I do not think metasurface is a good choice since the size of the device in this work is not small (>10 micrometers). Quite some works reported light-driven nanomotors in the past decade.
4. The authors also mentioned the material limitation in other strategies. However, when incorporating metasurface into the micromachines, material option is also limited to high refractive index materials/metals.
5. The authors emphasized the compatibility with stand photolithography for their fabrication techniques. However, laser printing was also used to create the pillars and caps. It seems that the fabrication is complicated and large area fabrication is still challenging.
6. The authors mentioned the laser-induced heating effect when they investigated the laser intensity dependency. I believe experimental measurement of the local temperature should be done and laser wavelength dependency should be given.

In summary, I believe the authors should better place their work along the line of light-driven micromachines and clearly point out the key challenge they have addressed in their work. More technical detail should be provided before the paper can be published.

Reviewer #2

(Remarks to the Author)

Please see the attached pdf.

Version 1:

Reviewer comments:

Reviewer #1

(Remarks to the Author)

The authors provided additional information to improve the manuscript and addressed part of my concerns in this revision. Still, I have some comments listed below:

- 1.As shown in Supplementary video 4, the geometry of single metarotor almost remains unchanged while the rotation speed decreases after long-term illumination. It seems that the rotor is not as stable as the authors claimed. More discussions are expected here and the rotor performance should be objectively described in the revised manuscript. In addition, SEM images of the rotor at initial stage and after continuous irradiation are preferred to better clarify the degradation or not.
- 2.Regarding maximum output torque or durability, the authors claim that light intensity can continuously increase the torque without a clear upper limit. However, high laser power may attribute to increased localized temperature, thermal fluidics or even bubbles to deteriorate the rotor performance, thus leading to an upper limitation of the maximum torque or durability.
- 3.Page 16, "in this word, tens of thousands of micromotors were fabricated on a 5 mm × 5 mm area". SEM images of the rotor array with thousands of rotors are required to show the scalability.
- 4.In this revision, the authors seem to avoid the discussion on power efficiency or energy conversion efficiency of the rotor, which is a key criterion for judging its performance. More in-depth and quantitative analysis are needed.
- 5.Picky, "operates effectivelu" should be "operates effectively".

Reviewer #2

(Remarks to the Author)

The authors have made extensive changes and properly addressed all my concerns. The only minor suggestion I have now is to provide a reference to the claim that their light power is "well within the safe thresholds for biological systems".

Version 2:

Reviewer comments:

Reviewer #1

(Remarks to the Author)

All my questions have been addressed. Now I can recommend this paper for publication.

Response to the Reviewers' reports

We thank the referees for their constructive comments and suggestions. We have carefully addressed each of their points in our response, making the necessary revisions to strengthen the manuscript.

Below, we provide a detailed point-by-point response to the referees' comments, including references to the specific pages in the revised manuscript where the changes have been incorporated.

RESPONSE TO REVIEWER #1

- *The work by Wang incorporated silicon metasurface to improve the light-matter interaction for light-driven micromotors. The authors have fabricated the meta-motor using photolithography and laser direct printing. They investigated the rotation velocity and direction at different laser intensities, geometries, and incident laser polarizations. They also demonstrated the gear trains powered by the metamotors, as well as the translation from rotational to linear motion.*

Our response: We thank the Referee for carefully reading our manuscript highlighting its main novelty points.

- *I have a couple of comments.*
 1. *The use of metasurface in light-driven micromachines is significant, but not new. A couple of works have been published in the past several years, such as Nature Nanotechnology volume 17, pages477-484 (2022), Nature Nanotechnology volume 16, pages970-974 (2021), etc. I noted that the authors have cited the second paper. I did not see the superiority of the designed metasurface/metamotors over those in these published works, regarding the functionalities and the efficiency of the micromachines. The authors should mention these relevant works and better place their work in the context.*

Our response: We thank the referee for pointing this out. We have revised the introduction of the manuscript and we now introduce different recent approaches to induce active motion via momentum transfer providing context for the current work.

We have revised the introduction adding the following material. The revised part now reads:

Recent advances in active matter have used unfocused light to propel microscopic vehicles employing plasmonic or dielectric metasurfaces that generate lateral optical forces through directional light scattering. For example, it has been shown that microvehicles with plasmonic or dielectric nanostructures arranged in a parallel pattern can move forward under linearly polarized light via linear momentum transfer (*Science Advance*, 2020, 6, eabc3726; *Nature Nanotechnology*, 2021, 16, 970-971) and can be steered using polarized light through spin angular momentum transfer (*Nature Nanotechnology*, 2021, 16, 970-971). Furthermore, it has also been shown that arranging the scatterers in a circular pattern enables rotation under linearly polarized light (*Science Advance*, 2020, 6, eabc3726; *Light: Science & Applications*, 2025, 14, 38). More advanced designs incorporate four individually addressable chiral plasmonic nanoantennas, allowing full 2D motion control through the application of dual-wavelength light. (*Nature Nanotechnology*, 2022, 17, 477-484).

In this work, we focus on utilizing metasurfaces for the fabrication of micromachines. Specifically, we employed the same metasurface as in Ref. [*Nature Nanotechnology*, 2021, 16, 970-971]. This metasurface is made of silicon, which is a fundamental material in the semiconductor industry and has lower optical absorption compared to gold nanoantennas (*Nature Nanotechnology*, 2022, 17, 477-484), which helps reduce the intensity of light required for the metasurface's operation. However, we modified its spatial distribution so that, while the original design moved linearly under linear polarization, our modification induces rotational motion. To clarify this modification, we have revised the manuscript to add following sentences to the results, which now read as:

The metasurface's unit cells, or *meta-atoms*, are composed of two asymmetric rectangular amorphous Si blocks, with dimensions $270\text{ nm} \times 200\text{ nm} \times 460\text{ nm}$ and $400\text{ nm} \times 200\text{ nm} \times 460\text{ nm}$, respectively, separated by a subwavelength gap of 50 nm to maximize the efficiency of the +1 light diffraction order relative to the 0 and -1 orders, based on the same design presented in [*Nature Nanotechnology*, 2021, 16, 970-971] (see more details in the Supplementary Note "Design principle of metasurface"). [...]

The metasurface of the metarotor in Figs. 1a-e comprises four segments, each with meta-atoms arranged in parallel but rotated by 90° relative to the adjacent segments. The direction of the resulting forces (F) acting on the metarotor under uniform linearly polarized light for each segment are schematically depicted using white arrows in Fig. 1g. [...] The forces (F) have a non-zero lever arm (r) relative to the common center of mass, generating a torque $\tau = rF$ that induces a counterclockwise rotation of the metarotor, as depicted by the black arrow in Fig. 1g. (more details in the Supplementary Note “Rotation mechanism of the motor”).

We have also included a discussion on potential improvements to the metasurface design to enhance the efficiency of the motors:

Alternative metasurface materials, such as TiO_2 , could extend the operational wavelength into the visible light region, simplifying optical calibration. These advancements could improve the system’s performance, adaptability, and applicability across diverse environments.

- 2. *The key to incorporate metasurface in light-driven micromachines is to introduce additional functionalities beyond those without metasurface, or to improve the optomechanic coupling efficiency of the devices. However, I did not see any discussion in both the main text and the SI on the analysis of light-metasurface interaction and generation of optical force. What is the design principle of the metasurface? Is it the optimized design? What is the optomechanic coupling efficiency? Is silicon the optimized materials? How about plasmonic metasurface which could improve the local electric field? These questions should be carefully addressed as they seem to me much more important than the control experiments given in the main text.*

Our response: In our study, we employed a similar metasurface design to our previous work (*Nature Nanotechnology*, 2021, 16, 970-974) without extensive optimization, as the key innovation of the current work lies in integrating the metasurface to shrink the on-chip light-driven micromachines—whereas our previous research primarily focused on single particles moving in solution.

Briefly, we fabricate the metasurface using a high refractive index material, amorphous silicon (a-Si), chosen for its low optical absorption, flexible light-field programming, and compatibility with large-scale semiconductor processing. The metasurface consists

of an asymmetric rectangular dimer structure, with its size and periodicity in two directions optimized through both FDTD simulation and experimentation to efficiently diffract light with $\theta \approx 57^\circ$ into +1 ($\sim 60\%$ intensity in transmission, $\sim 15\%$ in reflection) order while minimizing the -1 ($\sim 5\%$ intensity both in transmission and reflection) and 0 ($\sim 10\%$ intensity in transmission, $\sim 5\%$ in reflection)) orders along the long axis of the meta-atoms. Due to the conservation of linear optical momentum, directional light deflection generates a reaction force, $\sim 55\%$ of the total incident linear momentum along z-axis is converted into a reactive force in the positive x direction. The details are provided in the Methods and Supplementary Material of our previous work published in *Nature Nanotechnology*, 2021, 16, 970-974).

In contrast to previous work published in *Nature Nanotechnology*, 2021, 16, 970-974, where the particle embedded with the metasurface only performed linear motion under linear polarization, in the current work we embed four metasurface patches into the motor. Each patch contains meta-atoms arranged in parallel but rotated by 90° relative to adjacent segments, which have different force (F) and arms (r_0) directions. Based on the equation $\tau = 2r_0F$, this configuration generates a torque that drives the motor to rotate continuously in the counterclockwise direction.

Following the reviewer's suggestion and to improve clarity for readers, we have revised the article to include a discussion of the metasurfaces' design in the results, which reads now as:

The fabrication process involves four key steps. First (Fig. 1b), the metasurface is etched, optimized for operation in water under 1064 nm plane wave illumination. The metasurface's unit cells, or *meta-atoms*, are composed of two asymmetric rectangular Si blocks, with dimensions $270 \text{ nm} \times 200 \text{ nm} \times 460 \text{ nm}$ and $400 \text{ nm} \times 200 \text{ nm} \times 460 \text{ nm}$, respectively, separated by a subwavelength gap of 50 nm to maximize the efficiency of the +1 light diffraction order relative to the 0 and -1 orders, based on the design presented in Ref. [*Nature Nanotechnology*, 2021, 16, 970-974] (see more details in the Supplementary Note "Design principle of metasurface").

We have also added a Supplementary Note in the Supplementary Materials to elaborate on the metasurfaces used in our work and the principles of the rotation of motors. This new section of the Supplementary Materials reads:

Design principle of the metasurface

Here, we describe the rotation mechanism of the motor driven by the metasurface, which includes the design principle of the metasurface and the process of light momentum exchange.

For our motor, we adopted the metasurface design from Ref. [*Nature Nanotechnology*, 2021, 16, 970-974]. The metasurface operates based on a two-dimensional periodic structure with lattice constants Λ_x and Λ_y , embedded in a medium with refractive index n_b . According to Bloch's theorem, a normally incident plane wave is scattered into diffraction orders defined by the in-plane wavevector $\mathbf{k}_{\parallel} = \left(\frac{2\pi n}{\Lambda_x}, \frac{2\pi m}{\Lambda_y}\right)$. Only orders satisfying $|\mathbf{k}_{\parallel}| < \frac{2\pi n_b}{\lambda}$ can propagate. To enable propulsion along the x -direction while suppressing motion along the y -direction, the lattice is designed with $\Lambda_y < \frac{\lambda}{n_b} < \Lambda_x$. This configuration allows propagation in the $m = 0$ subspace but suppresses $m \neq 0$ orders. For our design, $\Lambda_x = 950$ nm and $\Lambda_y = 600$ nm, resulting in a diffraction angle of $\theta \approx 57^\circ$.

The metasurface unit cell consists of an asymmetric dimer nanoantenna (meta-atoms), comprising two rectangular Si blocks with dimensions 270 nm \times 200 nm \times 460 nm and 400 nm \times 200 nm \times 460 nm, separated by a 50 nm gap. This design maximizes the efficiency of the $+1$ diffraction order ($\sim 60\%$ intensity in transmission, $\sim 15\%$ in reflection) relative to the 0 ($\sim 10\%$ intensity in transmission, $\sim 5\%$ in reflection) and -1 ($\sim 5\%$ intensity in both transmission and reflection) orders along the long axis of the meta-atoms. The directional deflection of light generates a reaction force F_{opt} along the lattice period $\Lambda_x = 950$ nm due to the conservation of linear optical momentum, so that $\sim 55\%$ of the total incident linear momentum along z -axis is converted into a reactive force in the positive x -direction.

Rotation mechanism of the motor

The metasurface on our motor is divided into four segments, each containing meta-atoms arranged in parallel but rotated by 90° relative to the adjacent segments. Based on the theory reported in Ref. [*Light: Science & Applications*, 2025, 14, 38; *Nano Letters*, 2025, doi:10.1021/acs.nanolett.4c06410], the force generated by one of the segment metasurfaces on the motor can be expressed as

$$F = F_0 f(\varphi)$$

where F_0 is the radiation pressure force that the incident field would generate on a completely absorptive object with the same geometrical cross-section as the metasurface area. This force corresponds to 100% light deflection with a diffraction angle of $\theta = 90^\circ$. The value of F_0 can be calculated from the Minkowski momentum as

$$F_0 = \frac{n}{c_0} P_0$$

where n is the refractive index, P_0 is the optical power, and c_0 is the speed of light in vacuum.

Under linearly polarized light, as the motor rotates with a relative angle φ between one segment of the metasurface and the light polarization, the metasurface is only able to deflect a polarization-dependent fraction of the incident light momentum in the preferred direction. The function $f(\varphi)$ is used to calibrate this:

$$f(\varphi) = f_p \cos^2 \varphi + f_s \sin^2 \varphi$$

where

$$f_{p,s} = (T_{p,s}^{+1} - T_{p,s}^{-1} + R_{p,s}^{+1} - R_{p,s}^{-1}) \sin(\theta)$$

Here, $T_{p,s}^{+1}$ and $R_{p,s}^{+1}$ are the power diffraction efficiencies for transmission and reflection at different orders. The indices p and s represent polarization parallel or perpendicular to the plane of diffraction, respectively.

The torque generated by the metasurface to make the motor rotate can be expressed

as

$$\tau = r_0 F$$

where r_0 is the moment arm.

For the four-segment metasurface on the motor, with different angles between the metasurface and the light polarization, we have

$$\varphi_1 = \varphi, \quad \varphi_2 = \varphi + \frac{\pi}{2}, \quad \varphi_3 = \varphi + \pi, \quad \varphi_4 = \varphi + \frac{3\pi}{2}.$$

Thus, the total torque can be expressed as

$$\tau = 2r_0 \frac{n}{c_0} P_0 (f_p + f_s)$$

or equivalently,

$$\tau = 4r_0 \frac{n}{c_0} P_0 (T^{+1} - T^{-1} + R^{+1} - R^{-1}) \sin(\theta)$$

where T^{+1} , T^{-1} , R^{+1} , and R^{-1} represent the average transmission (T) and reflection (R) efficiencies of the diffracted light, with θ denoting the diffraction angle.

As the motor is working at low Reynolds numbers ($\text{Re} \sim 10^{-4}$), inertial effects are small. Furthermore, since the motor has a size of several micrometers, Brownian diffusion is hardly noticeable. Therefore, the angular velocity (ω) can be expressed as

$$\omega = \dot{\varphi}(t) = \frac{\tau(t)}{\gamma_r}$$

where γ_r is the rotational friction. The motor can be approximated as a thin disk, and the rotational friction can be estimated as $\gamma_r = \frac{32}{3}\eta r^3$ [*Applied Physics Letters*, 2001, 78, 547-549], where r is the radius of the motor, and η is the viscosity of water.

- 3. *The author claimed in the introduction along the line of miniaturization of micromotors. However, I do not think metasurface is a good choice since the size of the device in this work is not small (≈ 10 micrometers). Quite some works reported light-driven nanomotors in the past decade.*

Our response: The referee correctly points out that there are a range of smaller light-driven motors, which include nanoparticle and even individual molecules capable of

rotating upon light illumination (see for example *Nature*, 2006, 440, 163; *Nano Letters*, 2010, 10, 268-273; *Physical Review Letters*, 2018, 121, 033603). However, we would like to clarify that the focus of our work is not the miniaturization of individual motors but rather the miniaturization of functional mechanical machines—specifically, geared mechanisms. We define mechanical machines as consisting of at least two mechanical parts that interact with each other and form a coordinated, functional unit.

We should point out that while far-field driven nanomotors have indeed demonstrated remarkable progress in terms of individual motor miniaturization, they have largely been limited to isolated systems and have not yet achieved the integration of multiple mechanical components into complex mechanical machines. In contrast, batch-fabricated mechanical machines have been demonstrated with typical dimensions down to 50 μm . Even smaller machines, around 15 μm , can be achieved through piece-part assembly, though this approach lacks scalability.

Thus, while the size of our metasurface-driven motors is larger than some reported nanomotors (see Table S1), our work achieves a significant reduction in the size of functional mechanical systems, reaching the 10-micrometer scale. This represents a notable advancement compared to previously reported systems, which typically operate at the scale of hundreds of micrometers.

To clarify these points and include the above-mentioned definition of a mechanical machine, we have revised the introduction and added the following sentences:

While far-field approaches such as AC electric, magnetic, and light fields allow further miniaturization of individual micromotors (Table S1), they present their own limitations in achieving the integration of multiple motors into on-chip complex geared mechanical machines, which can be defined as at least two mechanical parts interacting with each other to form a cohesive unit capable of generating and transmitting work.

Additionally, we added Table S1, summarizing and comparing current mechanisms to drive micromotors, their dimensions, rotational speed and possibilities to couple them into geared mechanism to the supplementary information.

Table S1: Comparison of mechanisms to drive micromotors

Driving methods	Material of motors	Working medium	Dimension of individual motors	Reported maximum rotation speed	Addressable control	Coupling motors to functional microscopic geared mechanisms		Ref
						Size	Coupling methods	
Static	Si, SiO ₂ , Si ₃ N ₄	Air	120 μm diameter	500 rpm	NA (Difficult)		No	[1]
Static	Carbon Nanotubes	Air	2 μm in length	NA	NA (Difficult)		No	[2]
Static	SU8	oil	100 μm diameter	3000 rpm	NA (Difficult)		No	[3]
Static (Comb drive)	Si, SiO ₂	Air	50 μm diameter	~ 200000 rpm	NA (Difficult)	> 100 μm	Completely batch-fabricated	[4]
Ac electric	Au@Ni nanorods	DI water	10 μm in length	18000 rpm	NA (Difficult)		No	[5]
Ac electric	Au@SU8	DI water	~ 25 μm diameter	~ 23 rpm	NA (Difficult)		No	[6]
Magnetic	Fe ₂ O ₃ particles @resist	DI water	~ 30 μm diameter	~ 150 rpm	NA (Difficult)		No	[7]
Magnetic	NdFeB particles @resist	DI water	~ 100 μm diameter	~ 720 rpm	NA (Difficult)	~ 300 μm	Completely batch-fabricated	[8]
Chemical	Pt@resist	H ₂ O ₂ /H ₂ SO ₄ /DI water	120 μm diameter	60 rpm	NA (difficult)		No	[9]
Chemical	Pt particles@resist	H ₂ O ₂	~ 30 μm diameter	~ 160 rpm	NA (Difficult)	~ 50 μm	Completely batch-fabricated	[10]
Micro-organisms	SU8	Buffer	~ 15 μm diameter	~ 15 rpm	Yes		No	[11]
Optical (OET)	SU8	DI water	150 μm diameter	~ 83 rpm	Yes	> 300 μm	Piece-part assembly	[12]
Optical	Polymer (3D-shaped)	DI water	4.5 μm diameter	NA	Yes	~ 15 μm	Piece-part assembly	[13]
Optical (nanoantenna)	Au@resist	DI water	2.5 μm diameter	~ 190 rpm	Yes		No	[14]
Optical (metasurface)	Si, SiO ₂	DI water	8 μm diameter	~ 240 rpm	Yes		No	[15]
Optical (metasurface)	Si, SiO ₂ , SU8	DI water	~ 10 μm diameter	120 rpm	Yes	≤ 50 μm	Completely batch-fabricated	This work

Table notes: 1. Static (Combe drive): Microelectromechanical Systems (MEMS) linear actuators using electrostatic forces between conductive combs. 2. Optical (OET): OptoElectronic Tweezer. 3. The table column “Coupling motors to functional microscopic geared mechanisms” is split into “Size” and “Coupling methods.” “Size” indicates the overall dimensions of the microscopic geared mechanisms, while “Coupling methods” specifies how motors are connected to them.

- 4. *The authors also mentioned the material limitation in other strategies. However, when incorporating metasurface into the micromachines, material option is also limited to high refractive index materials/metals.*

Our response: We thank the Referee for pointing out that the respective sentence in the introduction was misleading. Indeed, incorporating metasurfaces into micromachines requires the use of high-refractive-index materials or metals to achieve optimal optical performance. What we intended to convey is that the material requirements for far-field-driven motors, such as those employing AC electric or magnetic fields, can restrict their integration into mechanical machines.

Methods using AC electric or magnetic fields require motors to be made from specific field-responsive materials—conductive materials for electric fields or magnetic materials for magnetic fields. These requirements not only limit material choices but also complicate integration with commercial semiconductor microfabrication techniques, making it challenging to combine multiple mechanical components into functional systems. In contrast, the materials used in metasurfaces, especially in our case (Silicon, SiO₂) are already standard in microfabrication processes and are fully compatible with photolithography, enabling seamless integration into existing manufacturing workflows.

While metasurfaces require high-refractive-index materials or metals for efficient light manipulation, these materials are widely available and can be optimized independently of other functional needs. This flexibility enables the design of complex mechanical systems. For example, the machine in Fig. 4 is challenging to fabricate using electric or magnetic fields, as field-responsive materials make it difficult to program the coupling of rotational and translational motion for simultaneous control. Importantly, the compatibility of metasurface materials with standard fabrication techniques facilitates

the seamless integration of multiple components, a significant advantage over other strategies that struggle with such integration.

To provide a clearer explanation, we have revised the Introduction as follows:

Methods employing AC electric or magnetic fields require field-responsive materials (e.g., conductive or magnetic materials), complicating integration into mechanical machines and making systems prone to external interference. Integrating multiple components further risks cross-interference, potentially degrading functionality.

- 5. *The authors emphasized the compatibility with stand photolithography for their fabrication techniques. However, laser printing was also used to create the pillars and caps. It seems that fabrication is complicated and large area fabrication is still challenging.*

Our response: In our study, direct laser writing (maskless aligner) was used to expose the SU-8 photoresist to create the pillars and caps. Since we only write 2D structures, unlike 2-photon lithography, direct laser writing is a comparably fast writing technique. We chose direct laser writing primarily for its flexibility, allowing us to rapidly iterate designs and optimize manufacturing parameters—a key part of this study. Further upscaling and large-area production would involve employing established fabrication techniques such as deep ultraviolet (DUV) lithography and photolithography.

To address the Referee’s concern and provide a clearer explanation, we have added a section titled “Further Optimization of the Fabrication Process” in the Methods section of the revised manuscript:

Further Optimization of the Fabrication Process

In this work, tens of thousands of micromotors were fabricated on a 5 mm × 5 mm area. The metasurface was created using electron beam lithography (EBL), while direct laser writing was used for the micromotor pillars and caps. These lab-based techniques offer design flexibility and rapid iteration but are highly time-consuming for wafer-scale manufacturing, requiring days to complete. Foundry-compatible alternatives,

such as deep ultraviolet (DUV) lithography and nanoimprinting [*Light: Advanced Manufacturing*, 2025, 5, 117-132; *Nature Material*, 2023, 22, 474-481.], can replace EBL for metasurface fabrication, while lithography and imprinting techniques can substitute direct laser writing for pillar and cap fabrication [*Electrophoresis*, 2007, 28, 4539-4551], enabling cost-effective scalability.

- 6. *The authors mentioned the laser-induced heating effect when they investigated the laser intensity dependency. I believe experimental measurement of the local temperature should be done and laser wavelength dependency should be given.*

Our response: The focus of our current work is to utilize micromotors for the construction of micromachines rather than to develop new motor principles. Given the challenges associated with accurately characterizing the local temperature (see below), we rely on numerical simulations (Supplementary Fig. 3) to estimate the temperature distribution around the metasurface. In the revised version of the manuscript, we have expanded these simulations to specifically address the concerns raised by Reviewers 1 and 2. From the temperature, we calculate the viscosity of water, which allows us to determine the rotational friction coefficient of the motor. Based on the torque generated by the metasurface (detailed in Supplementary Note: Rotation mechanism of motor), we obtain the angular velocity of the motor as a function of light intensity (Supplementary Fig. 6, as shown below). The simulation results show that the relationship between angular velocity and light intensity is superlinear rather than linear, which aligns well with the experimental results. This agreement confirms that increasing light intensity induces temperature changes, which in turn affect the angular velocity trends of the motor. This finding is further supported by a recently published study (*Light: Science and Applications*, 2025, 14, 38).

We added the following figure to the supplementary material to support our conclusions:

FIG. S6: **Calculated optical torque (τ), rotational friction coefficient (γ_r), and angular velocity (ω) of four metarotors with different numbers of meta-atoms (22, 29, 36, 55) under varying illuminated light intensities. **a** Calculated optical torque of four metarotors under different light intensities based on the model in the Supplementary Note. The optical torque ($\tau = rF$) and the optical force (F) increase linearly with light intensity. **b** Calculated rotational friction coefficient for four metarotors at different light intensities. Higher light intensity results in a smaller rotational friction coefficient. **c** Angular velocity ($\omega = \tau/\gamma_r$) for the four metarotors. ω increases superlinearly with varying intensities, consistent with experimental trends.**

Challenges in Local Temperature Measurement:

We agree that direct temperature measurements near our microgears would provide experimental validation for our hypothesis that the observed velocity increase results from a local temperature rise. Various experimental techniques, such as fluorophores, fiber sensors, thermal cameras and Raman spectroscopy, could in principle be used for temperature measurements. While these methods have successfully measured local temperatures in other systems, their implementation in our setup presents several challenges, which we will discuss individually.

Fluorescence: One established approach for such measurements involves quantifying shifts in fluorophore intensity, a technique previously used for thermal imaging of plasmonic nanostructures [*Nature Communication*, 2015, 6, 7915]. In these studies, fluorophores were embedded in a PMMA film surrounding the nanostructures and analyzed using a custom-built confocal microscope. While, in principle, our experimental setup could be adapted to operate similarly, significant technical challenges arise in our specific case. The key limitation is that, unlike in previous work where

fluorophores were immobilized in a solid PMMA film, our system requires fluorophores to be dissolved in the aqueous phase. This fundamental difference introduces several complications:

The micromotor is positioned close to the substrate, and the laser-induced temperature increase generates thermal convection in the surrounding fluid. This convection, combined with the motor's rotation, creates a complex flow field that can influence the distribution of fluorescent molecules in fluorescence-based thermometry or distort the thermal radiation profile in infrared (IR) thermography. In contrast, when measuring baselines in static systems such as pure glass chambers, this convection flow would be absent. These effects introduce significant challenges in obtaining accurate and reliable temperature measurements, as they can lead to artifacts that obscure the true thermal distribution.

The localized heating effect occurs at the microscale, with a highly dynamic temperature distribution influenced by the motor's rotation and the resulting fluid flow. Capturing these rapid, localized temperature changes with the required spatial and temporal resolution is highly challenging and requires careful experimental design using current techniques.

Fiber sensors: The extremely small scale of the metasurface and the challenges of positioning commercial fiber temperature sensors—typically millimeter-sized—in such a confined and dynamic environment make direct experimental measurement of the local temperature highly challenging. Introducing sensors could disrupt fluid flow, alter the thermal profile, or interfere with micromotor rotation, potentially leading to unreliable results.

Thermal Camera: We use a 1064 nm laser to induce micromotor rotation. Temperature measurement techniques, such as infrared thermography, would be significantly affected by the IR laser itself, as its energy could interfere with thermal radiation signals. This interference may lead to inaccurate temperature readings. Additionally, thermal imaging has poor spatial resolution and requires specialized IR optics, further limiting its suitability for precise temperature measurements in our setup.

Raman spectroscopy: Although non-contact temperature measurement methods based on the Raman scattering effect are available, the material of the metasurface

is amorphous silicon rather than polycrystalline silicon. Due to the absence of lattice periodicity, the Raman scattering signal of amorphous silicon is relatively broad and diffuse, making it challenging to extract temperature information from the Raman spectrum and obtain a clear and accurate signal.

Laser Wavelength Dependency and Temperature Change:

In this study, we optimized the metasurface design for a 1064 nm laser and did not investigate the effects of different wavelengths. Since silicon's absorption increases exponentially at shorter wavelengths, we expect significantly stronger heating with visible light compared to the infrared light used here. Conversely, if the design is optimized for longer wavelengths, heating due to light absorption by the metasurface becomes negligible but the absorption of water starts becoming relevant.

- *In summary, I believe the authors should better place their work along the line of light-driven micromachines and clearly point out the key challenge they have addressed in their work. More technical detail should be provided before the paper can be published.*

Our response: We thank Referee's insightful feedback and constructive suggestions, and we have revised the article in response to these suggestions. We hope these revisions enhance the clarity and impact of our work.

RESPONSE TO REVIEWER #2

- *This manuscript builds upon prior work (Nature Nanotechnology 16, 970–974 (2021)) by further extending the application of programmable optical metasurfaces to microscopic gear mechanisms. Through several micro-mechanical models, the study effectively demonstrates the gear system’s high motion performance, precise controllability, and scalability, creating a practical and multifunctional platform for micro- and nanoscale mechanical systems. Overall, the work presents an innovative approach to microscale gear mechanisms with strong potential. However, I recommend major revisions to address my concerns listed below to improve the manuscript’s impact, clarity, and broader relevance*

Our response: We thank the Referee for carefully evaluating our manuscript and for appreciating its novelty. We address the Referee’s concerns below.

- *1. Practical limitation*

High Light Intensity Requirement

The system requires a light intensity of approximately $35 \mu\text{W } \mu\text{m}^{-2}$ to achieve typical rotational speeds. This necessitates complex external optical setups, which could limit real-world applications. A discussion on potential power reduction strategies or trade-offs would be valuable. In addition, is this power level safe for biomedical applications?

Our response: We thank the Referee for raising this important point regarding the light intensity requirements and their potential impact on real-world applications. We appreciate the opportunity to clarify these aspects.

The system utilizes a 1064 nm laser, which is considered biologically safer due to its lower absorption by water and biological tissues compared to shorter wavelengths. This minimizes the potential for damage, making it suitable for biomedical applications.

The reported light intensity of $35 \mu\text{W } \mu\text{m}^{-2}$ is based on a spot size of approximately 300 μm . However, since the driving gear, which powers the entire system, has a diameter of only 16 μm , the light can be precisely focused on this smaller area. Consequently, the total power required is only about 7 mW, a relatively low value that remains well within safe limits for biomedical use.

Furthermore, the light is directed solely to the driving gear, which subsequently transfers motion to the other passive structures. These passive structures, rather than the light itself, would interact with the biological samples. This design ensures that biological samples are not directly exposed to the laser, thereby mitigating concerns regarding potential light-induced damage.

Based on these considerations, we have clarified this aspect in the discussion of our manuscript:

By using light as a widely available and biocompatible energy source, these micromotors are well-suited for manipulating biological matter, including bacteria and cells. The system employs a 1064 nm laser, which minimizes damage to biological samples due to its low absorption by water and tissues. The light can be focused from a large area onto the small driving gear, operating at a low power requirement of just a few mW, which remains well within the safe thresholds for biological systems. Importantly, the light can be selectively directed to the driving gear, allowing it to mechanically actuate passive structures without directly exposing biological samples to the light source. This non-toxic, indirect energy delivery mechanism broadens the applications of our light-driven micromotors and metamachines in biomedical environments.

- *Limited Motion Adjustability*

The reliance on pre-designed optical metasurfaces somewhat restricts the flexibility in dynamically modifying gear motion modes. The manuscript should acknowledge this limitation and explore possible workarounds, such as reconfigurable metasurfaces or adaptive optics.

Our response: We fully agree with the Reviewer and we are currently exploring multiple exciting ways to dynamically modify the gear motion as a response to external stimuli. For example, we are currently explore the integration of phase-transition materials (e.g., VO₂) into the metasurface design, enabling real-time reconfiguration of optical properties in response to external stimuli such as temperature, electric fields, or light. Moreover, we are exploring the use of two-photon lithography to connect motors with flexible, stimuli-responsive polymers, allowing to dynamically change the shape and configuration of the gears. While both strategies are highly promising, they

each add an additional layer of complexity and thus we feel that they are beyond the scope of this publication.

Following the Reviewer's suggestion to acknowledge the current limitations and highlight the possibilities for more complex, future micromachines, we added the following paragraph as an outlook in the discussion section:

The use of phase-transition materials [*Nature Nanotechnology*, 2021, 16, 661-666] (e.g., VO₂) could be integrated into the metasurface design, enabling real-time reconfiguration of optical properties in response to external stimuli such as temperature, electric fields, or light [*Advanced Functional Materials*, 2019, 29, 1806692]. This would address the current limitation of relying on pre-designed metasurfaces, which restricts dynamic motion adjustability. Additionally, adaptive optics, including deformable mirrors or spatial light modulators, could enhance flexibility by enabling precise wavefront correction and dynamic light modulation.

- *Scalability and Large-Scale Fabrication*

While the manuscript highlights CMOS compatibility, it lacks quantitative discussion on scalability. Specifically: How many micromotors can be integrated on a single chip?

Our response: On a single chip, the number of integrated micromotors depends on the unit cell size and the available chip area. In our process, we begin with 4-inch wafers, which are diced into chips measuring 13 mm × 13 mm. Electron beam lithography is performed on these chips; however, only a 5 mm × 5 mm area is patterned for micromotor fabrication, with the remainder reserved for mounting a PDMS spacer to form a water-filled chamber. Within this 5 mm × 5 mm area, micromotors with diameters of approximately 16 μm and a spacing of 16 μm are fabricated, yielding tens of thousands of devices per chip.

We added the following sentence to the results of the manuscript, which reads as:

Using this approach, tens of thousands of micromotors can be fabricated within a 5 mm × 5 mm area on a single chip. Moreover, the fabrication process can be scaled up to the wafer level.

Are there fabrication constraints that could hinder large-scale production?

Our response: The critical aspect in the fabrication process is the formation of caps on top of the pillars within a suspended three-dimensional structure. The construction of such a structure using a surface-fabrication method presents significant challenges. To achieve this, we employ a sacrificial layer technique utilizing AZ positive photoresist as the sacrificial material. However, this approach introduces fabrication difficulties, including non-uniform photoresist spin coating and cap collapse during the photoresist removal process, which can hinder the scalability of motor fabrication. To mitigate these limitations, an imprinting method can serve as an alternative to the sacrificial layer approach. Furthermore, to enable rapid and flexible design iterations, we utilize maskless electron beam lithography and laser writing. While these techniques allow for large-scale fabrication, they are often time-intensive. For industrial-scale manufacturing, these methods can be seamlessly replaced with mask lithography.

To clarify this point, we added the following paragraph to the Methods:

Further Optimization of the Fabrication Process

In this work, tens of thousands of micromotors were fabricated on a $5\text{ mm} \times 5\text{ mm}$ area. The metasurface was created using electron beam lithography (EBL), while direct laser writing was used for the micromotor pillars and caps. These lab-based techniques offer design flexibility and rapid iteration but are highly time-consuming for wafer-scale manufacturing, requiring days to complete. Foundry-compatible alternatives, such as deep ultraviolet (DUV) lithography and nanoimprinting [*Light: Advanced Manufacturing*, 2025, 5, 117-132; *Nature Material*, 2023, 22, 474-481], can replace EBL for metasurface fabrication, while lithography and imprinting techniques can substitute direct laser writing for pillar and cap fabrication [*Electrophoresis*, 2007, 28, 4539-4551], enabling cost-effective scalability.

How does optical alignment complexity scale as the number of micromotors increases?

The micromotors we developed are driven by a planar light wave: a 1064 nm laser is weakly focused to a spot size of $300\text{ }\mu\text{m}$ to provide the necessary light intensity for rotation. Under illumination, multiple micromotors rotate stably and simultaneously.

By moving the stage, the other motors on the chip can also move. We have revised the manuscript to add the following sentence to results and included a new Supplementary Video (SI Video 7) to demonstrate this capability.

Multiple metarotors rotate stably and simultaneously under uniform illumination (Supplementary Video 7). Increasing the spot size and/or the laser power enables control of more micromotors over a larger area.

- 2. *Comparison with Competing Technologies*

The manuscript introduces metasurface-powered gears as a superior alternative to existing methods, but the justification remains qualitative. A more quantitative comparison with other techniques is necessary. An additional class of work has explored electrostatic-driven micromechanical gears (e.g., <https://doi.org/10.3311/PPch.10274>). These do not require complex electrical connections around the gears, enable large-scale parallelization, can be fabricated using standard photolithography, and do not require intricate metasurface designs or external optical control. I encourage the authors to consider: How does the power efficiency of metasurface-driven gears compare to electrostatic gears? Can the scalability and manufacturability of these systems be directly compared?

*Other studies, such as Catchmark, Jeffrey M.; Subramanian, Shyamala; Sen, Ayushman, Directed rotational motion of microscale objects using interfacial tension gradients continually generated via catalytic reactions, *Small* (2005), 1 (2), 202-206, have demonstrated chemically powered microgears. These achieve similar functionalities and do not require external optical control*

Finally, the manuscript claims that optoelectronic tweezers (OET) “lack flexibility,” making them inferior to metasurface-driven systems. However, OET demonstrations show similar functionalities to those presented here. Could the authors provide a head-to-head, quantitative comparison with OET? Where does the proposed technique outperform OET, and in which aspects might OET remain more practical?

A comparative table, and a dedicated discussion, summarizing advantages and trade-offs across these technologies would significantly improve the manuscript’s clarity and scientific positioning. I do not expect the current technique to win it all, but it would

Table S1: Comparison of mechanisms to drive micromotors

Driving methods	Material of motors	Medium	Dimension of individual motors	Reported maximum rotation speed	Addressable control	Coupling motors to functional microscopic geared mechanisms		Ref
						Size	Coupling methods	
Static	Si, SiO ₂ , Si ₃ N ₄	Air	120 μm diameter	500 rpm	NA (Difficult)		No	[1]
Static	Carbon Nanotubes	Air	2 μm in length	NA	NA (Difficult)		No	[2]
Static	SU8	oil	100 μm diameter	3000 rpm	NA (Difficult)		No	[3]
Static (Comb drive)	Si, SiO ₂	Air	50 μm diameter	~ 200000 rpm	NA (Difficult)	> 100 μm	Completely batch-fabricated	[4]
Ac electric	Au@Ni nanorods	DI water	10 μm in length	18000 rpm	NA (Difficult)		No	[5]
Ac electric	Au@SU8	DI water	~ 25 μm diameter	~ 23 rpm	NA (Difficult)		No	[6]
Magnetic	Fe ₂ O ₃ particles @resist	DI water	~ 30 μm diameter	~ 150 rpm	NA (Difficult)		No	[7]
Magnetic	NdFeB particles @resist	DI water	~ 100 μm diameter	~ 720 rpm	NA (Difficult)	~ 300 μm	Completely batch-fabricated	[8]
Chemical	Pt@resist	H ₂ O ₂ /H ₂ SO ₄ /DI water	120 μm diameter	60 rpm	NA (difficult)		No	[9]
Chemical	Pt particles@resist	H ₂ O ₂	~ 30 μm diameter	~ 160 rpm	NA (Difficult)	~ 50 μm	Completely batch-fabricated	[10]
Micro-organisms	SU8	Buffer	~ 15 μm diameter	~ 15 rpm	Yes		No	[11]
Optical (OET)	SU8	DI water	150 μm diameter	~ 83 rpm	Yes	> 300 μm	Piece-part assembly	[12]
Optical	Polymer (3D-shaped)	DI water	4.5 μm diameter	NA	Yes	~ 15 μm	Piece-part assembly	[13]
Optical (nanoantenna)	Au@resist	DI water	2.5 μm diameter	~ 190 rpm	Yes		No	[14]
Optical (metasurface)	Si, SiO ₂	DI water	8 μm diameter	~ 240 rpm	Yes		No	[15]
Optical (metasurface)	Si, SiO ₂ , SU8	DI water	~ 10 μm diameter	120 rpm	Yes	≤ 50 μm	Completely batch-fabricated	This work

be very useful to see where it wins and where other technologies might be more useful.

Our response: We appreciate the Referee’s comment. Following the Referee’s suggestion, we have added a table in the Supplementary material highlighting different driving mechanisms for both individual motors and functional microscopic geared systems (which is the specific focus of this work):

Table notes: 1. Static (Combe drive): Microelectromechanical Systems (MEMS) linear actuators using electrostatic forces between conductive combs. 2. Optical (OET): OptoElectronic Tweezer. 3. The table column “Coupling motors to functional microscopic geared mechanisms” is split into “Size” and “Coupling methods.” “Size” indicates the overall dimensions of the microscopic geared mechanisms, while “Coupling methods” specifies how motors are connected to them.

Additionally, we have expanded our discussion in the Supplementary Materials to outline the advantages and limitations of various motor types, clearly demonstrating why our approach stands out as one of the most effective and scalable strategies for developing functional geared mechanisms. This additional section read as:

Comparison of mechanisms to drive micromotors

In Table S1, we systematically present various methods for constructing micromotors comparing their potential for enabling functional machines.

Although mechanisms such as electrostatic, alternating electric, magnetic fields, and chemical propulsion can be applied to drive individual micromotors and demonstrate excellent performance in terms of smaller dimensions, high rotational speeds, and efficiency, they still face numerous challenges for the construction of functional microscopic geared mechanisms.

Electrostatically driven motors can operate in air, enabling high rotational speeds and conversion efficiency (*Sensors and actuators*, 1989, 20, 41; *Nature*, 2003, 424, 408). However, they require the placement of driving electrodes near the motor, which limits the integration of other components and hinders the construction of complex machines. While electrostatically driven comb structures can be used for integration, their dimensions typically exceed 100 μm (*IEEE Electron Device Letters*, 1996, 17, 366), and scalable manipulation is difficult to achieve. Some methods can avoid the construction of external electrodes and do not require the fabrication of complex structures (*Pe-*

riodica Polytechnica Chemical Engineering, 2017, 61, 15), but integrating materials with different dielectric constants still presents challenges in alignment and material integration during the fabrication process.

AC electric (*Nature Communications*, 2014, 5, 3632; *Advance Functional Materials*, 2018, 28, 1803465) and magnetic field (*Nano Letters*, 2021, 21, 1628; *Nature Communications*, 2022, 13, 2016) driven systems encounter similar issues. Although magnetic machines can be fabricated by selectively embedding magnetic particles into the motor, this approach necessitates multiple processing and magnetization steps, leading to complex procedures, prolonged processing times, and final dimensions typically exceeding 100 μm , which hinders integration and large-scale production.

Chemically driven motors exhibit high conversion efficiency and resemble natural biological motors (*Small*, 2005, 1, 202; *Nano Letters*, 2024, 24, 3176). However, they are constrained by specific operating conditions and normally rely on bubble generation via chemical reactions for propulsion. This dependency can lead to interference between different machines and negatively impact functional stability. Microorganism-driven motors face stability issues (*Nature Communications*, 2017, 8, 15974).

Optical driving methods, such as optoelectronic tweezers (OET) (*Nature Communications*, 2021, 12, 5349), rely on light-modulating devices like digital micromirrors, whose resolution and response speed limit the size of motors and machines (typically on the order of hundreds of micrometers) and result in slower motion speeds. Currently, OET primarily uses piece-part assembly for machine integration rather than fully batch fabrication, limiting scalability. However, it offers flexibility and high-throughput advantages, making it well-suited for biological applications. Other optical methods, such as optical tweezers, can enable motor miniaturization and machine construction (*Applied Physics Letters*, 2001, 78, 249). However, they still rely on piece-part assembly, which makes large-scale manufacturing challenging. Moreover, these methods require focused light fields, limiting the potential for large-scale control.

Metasurface technology presents a novel solution to these challenges by enabling completely batch-fabricated production, addressable control, and scalable manipulation (*Nature Nanotechnology*, 2022, 17, 477; *Light: Science and Applications*, 2025, 14, 38). However, existing research has largely focused on constructing freely rotating

motors in solution, rather than on functional micro- and nanoscale machines. This work overcomes these limitations, successfully achieving scalable manufacturing and integration of micro- and nanoscale machines.

- *3. Theoretical Predictions of Force and Torque*

The manuscript lacks a theoretical model or numerical simulation for the expected force and torque generated by the metasurface-driven micromotors. Instead, it relies on experimental measurements and qualitative analysis. I wonder if the authors can derive force and torque estimates based on optical momentum transfer and fluid dynamics? Would finite element modeling (FEM) or a simplified analytical model help predict performance under different conditions?

Our response: Following the Referee’s suggestion, we have now added a simplified analytical model based on light-momentum transfer to estimate the torque required to rotate the motor. Using this model, we calculated the motor torque, τ_{opt} , under varying light intensities based on the optical force (F) applied to the motor and the lever arm (r_0) of the force, given by $\tau = r_0 F$. The force is derived from the optical momentum imparted by the embedded metasurface, which maximizes the efficiency of the +1 light diffraction order relative to the 0 and -1 orders. Based on this, the torque can be expressed as

$$\tau = 4r_0 \frac{n}{c_0} P_0 (T^{+1} - T^{-1} + R^{+1} - R^{-1}) \sin(\theta)$$

where n is the refractive index of silicon, c_0 is the speed of light, T^{+1} , T^{-1} , R^{+1} , and R^{-1} represent the average transmission and reflection efficiencies of the diffracted light of +1 and -1 orders, and θ is the diffraction angle. We can obtain these values from the experimental data.

For more details, we have revised the article by adding a Supplementary Note about the rotation mechanism of the motor:

Rotation mechanism of the motor

The metasurface on our motor is divided into four segments, each containing meta-atoms arranged in parallel but rotated by 90° relative to the adjacent segments. Based on the same calculation process shown in [*Light: Science & Applications*, 2025, 14, 38; *Nano Letters*, 2025], the force generated by one of the segment metasurfaces on the motor can be expressed as

$$F = F_0 f(\varphi)$$

Here, F_0 is the radiation pressure force that the incident field would generate on a completely absorptive object with the same geometrical cross-section as the metasurface area. This force represents 100% light deflection with a diffraction angle of $\theta = 90^\circ$. The value of F_0 can be calculated from the Minkowski momentum as

$$F_0 = \frac{n}{c_0} P_0$$

where n is the refractive index, P_0 is the power, and c_0 is the speed of light in vacuum.

Under linear polarization light, as the motor rotates with a relative angle φ between one segment of the metasurface and the light polarization, the metasurface is only able to deflect a polarization-dependent fraction of the incident light momentum in the preferred direction. The function $f(\varphi)$ is used to calibrate this:

$$f(\varphi) = f_p \cos^2 \varphi + f_s \sin^2 \varphi$$

where

$$f_{p,s} = (T_{p,s}^{+1} - T_{p,s}^{-1} + R_{p,s}^{+1} - R_{p,s}^{-1}) \sin(\theta)$$

where $T_{p,s}^{+1}$ and $R_{p,s}^{+1}$ are the power diffraction efficiencies for transmission and reflection at different orders. The indices p and s represent polarization within or perpendicular to the plane of diffraction.

The torque generated by the metasurface to make the motor rotate can be expressed as

$$\tau = r_0 F$$

where r_0 is the moment arm.

For the four-segment metasurface on the motor, with different angles between the metasurface and the light polarization, we have

$$\varphi_1 = \varphi, \quad \varphi_2 = \varphi + \frac{\pi}{2}, \quad \varphi_3 = \varphi + \pi, \quad \varphi_4 = \varphi + \frac{3\pi}{2}.$$

The total torque can be expressed as

$$\tau = 2r_0 \frac{n}{c_0} P_0 (f_p + f_s)$$

or equivalently,

$$\tau = 4r_0 \frac{n}{c_0} P_0 (T^{+1} - T^{-1} + R^{+1} - R^{-1}) \sin(\theta)$$

where T^{+1} , T^{-1} , R^{+1} , and R^{-1} represent the average transmission (T) and reflection (R) efficiencies of the diffracted light, with θ denoting the diffraction angle.

As the motor is working at low Reynolds numbers ($Re \sim 10^{-4}$), inertial effects are small and, as the motor size is on the order of several micrometers, rotational Brownian diffusion is hardly noticeable, the angular velocity (ω) can be expressed as

$$\omega = \dot{\varphi}(t) = \frac{\tau(t)}{\gamma_r}$$

where γ_r is the rotational friction. The motor can be approximated as a thin disk and the rotational friction can be estimated as $\gamma_r = \frac{32}{3}\eta r^3$ [*Applied Physics Letters*, 2001, 78, 547-549], where r is the radius of the motor, and η is the viscosity of water.

When we have the torque and viscosities of water at different temperatures as a function of intensity, we can obtain the angular velocity (ω) of the motor using the relation:

$$\omega = \frac{\tau}{\gamma_r}$$

While a rigorous calculation of the optical force and torque would require surface integration of the stress tensor around the object, our model still yields accurate results within the scope of the current study. These theoretical results are consistent with the experimental data.

We have revised the manuscript to include this result as Supplementary Figures 5-6, which illustrates the calculated torque of the motor with different meta-atoms under varying light intensities.

FIG. S5: Calculated optical torque (τ), rotational friction coefficient (γ_r), and angular velocity (ω) for four metarotors with varying meta-atom numbers under different light intensities. **a** Calculated optical torque for motors with 22, 29, 36, and 55 metaatoms per metasurface section at light intensities of $12.7 \mu\text{W } \mu\text{m}^{-2}$, $30.3 \mu\text{W } \mu\text{m}^{-2}$, $48.0 \mu\text{W } \mu\text{m}^{-2}$, $70.8 \mu\text{W } \mu\text{m}^{-2}$, and $88.5 \mu\text{W } \mu\text{m}^{-2}$ based on the model presented in the Supplementary Note. The torque ($\tau = rF$) increases sublinearly with the number of meta-atoms across all intensities due to varying force arm (r) for four meta-atoms, while the optical force (F) increases linearly. **b** Rotational friction coefficient for the same metarotors. At lower intensities ($12.7 \mu\text{W } \mu\text{m}^{-2}$, $30.3 \mu\text{W } \mu\text{m}^{-2}$, $48.0 \mu\text{W } \mu\text{m}^{-2}$), γ_r scales nearly linearly with metaatom number. At higher intensities ($70.8 \mu\text{W } \mu\text{m}^{-2}$, $88.5 \mu\text{W } \mu\text{m}^{-2}$), γ_r decreases sublinearly. **c** Angular velocity ($\omega = \tau/\gamma_r$) for the four metarotors. ω increases sublinearly with meta-atom number at all intensities, consistent with experimental trends.

FIG. S6: Calculated optical torque (τ), rotational friction coefficient (γ_r), and angular velocity (ω) of four metarotors with different numbers of meta-atoms (22, 29, 36, 55) under varying illuminated light intensities. **a** Calculated optical torque of four metarotors under different light intensities based on the model in the Supplementary Note. The optical torque ($\tau = rF$) increases linearly as the optical force (F) increases linearly with light intensity. **b** Calculated rotational friction coefficient for four metarotors at different light intensities. Higher light intensity results in a smaller rotational friction coefficient. **c** Angular velocity ($\omega = \tau/\gamma_r$) for the four metarotors. ω increases superlinearly with varying intensities, consistent with experimental trends.

- 4. *Additional Recommendations*

1. *The manuscript states that increasing the number of meta-atoms raises the local temperature, reducing viscous drag. This should, in principle, lead to a superlinear increase in rotational speed, rather than the observed sublinear trend. However, the manuscript attributes the slowdown to localized heating effects. Can the authors clarify what specific factors (e.g., light absorption saturation, thermal expansion, changes in optical torque efficiency) contribute to this behavior, and whether they have been quantitatively modeled?*

Our response: We thank the Referee for pointing this out. We agree that as the number of meta-atoms increases, the torque τ grows, and the motor's rotational friction coefficient γ_r decreases due to higher temperatures and lower viscosity under illumination, leading to a larger angular velocity ω as $\omega = \frac{\tau}{\gamma_r}$. However, τ does not scale linearly with the number of metaatoms because $\tau = r_0 F$, where the optical force (F) scales linearly with the number of meta-atoms, but the force arm (r) depends on their spatial distribution on the motor.

To clarify this relationship, we derived the dependence of τ on the number of meta-atoms using the analytical model presented in the newly added Supplementary Note "Rotation mechanism of the motor". This analysis confirms a sublinear trend. Additionally, we calculated the rotational friction coefficient using data from Supplementary Figure 4, which further supports the sublinear trend in angular velocity, as illustrated in the figure below. To enhance clarity, we have revised the manuscript and included this discussion in the Supplementary Note, along with Supplementary Figure 5 for further details. See these changes in the answer to the previous point.

- 2. *Additional schematic illustrations marking the gap position and probability distribution on Fig. 1f/g would improve reader comprehension.*

Our response: Following the referee's suggestion, we have modified Fig. 1k/l by inserting schematic illustrations marking the gap position and probability distribution within the figure. The revised figure is reproduced below.

- 3. In Supplementary video 11, indicating the number of metasurfaces incorporated into each of the three racks would enhance clarity.

Our response: We thank the Referee for this suggestion and we have added the number of metasurfaces incorporated into each of the three racks in Video 11, with white text on top of the racks.

- 4. While the manuscript discusses metasurface quantity and light intensity effects on speed, it does not address maximum output torque or durability. Referencing Refs. 24 and 25, which analyze similar micromechanical systems, would strengthen the discussion.

Our response: Based on the analytical model, which has been newly added to the Supplementary Materials (Supplementary Note “Rotation mechanism of the motor”), the magnitude of the torque (τ) depends on the optical force (F) and the lever arm (r_0) of the force, as $\tau = r_0 F$. As light intensity increases, the optical force also increases,

resulting in a larger torque that induces the motor's rotation. However, revealing the maximum torque is challenging, as the light intensity can continuously increase without a clear upper limit.

Here, we calculate and show the torque based on the maximum light intensity of $88.5 \mu\text{W} \mu\text{m}^{-2}$ as presented in the manuscript. The torque can be obtained from the experimental results, where the angular velocity (ω) at $88.5 \mu\text{W} \mu\text{m}^{-2}$ is measured. Since we are operating at low Reynolds numbers, where viscous forces dominate and inertial effects are negligible, the torque can be calculated using the relation $\tau = \omega\gamma_r$, where γ_r is the rotational friction coefficient of the motor. The calculated value of the torque is $36 \text{ pN} \cdot \mu\text{m}$. To clarify this, we have revised the manuscript as follows:

The maximum calculated torque of the motor is $36 \text{ pN} \cdot \mu\text{m}$ based on $\tau = \omega\gamma_r$ under the intensity of $88.5 \mu\text{W} \mu\text{m}^{-2}$. At higher intensity, both the angular velocity and torque increase (more details are provided in the Supplementary note "Rotation mechanism of the motor")

The motor's durability is influenced by hydrodynamic interactions between the motor, pillar, and substrate in an aqueous solution. However, evaporation of the solution affects its rotation. The chip is currently packaged in a PDMS chamber, which provides insufficient sealing. Our measurements show that the motor remains stable for 11 hours under continuous illumination but typically fails after about a day due to evaporation. Improving packaging to prevent evaporation or implementing a recirculation system could significantly extend the motor's lifetime. During this process, we have not seen any significant degradation of the material. After the liquid evaporated and the chip was left for half a year, water was re-injected, and the motor still functioned.

We have added a new Supplementary Video showing that the motor can rotate continuously for around 11 hours and have revised the manuscript to include this information in the results of manuscript, which reads:

Regarding motor durability, although the chip is currently not optimally packaged to ensure long-term stability of the sample, the motor remains stable and operates effectively under illumination for eleven hours (Supplementary Video 4), with no observed degradation during this period.

- 5. *The authors suggest applying this gear mechanism to mirror actuation. How does its durability and precision compare to existing DMD-based electrostatic actuators?*

Our response: We thank the Referee for this question regarding the comparison of our gear mechanism for mirror actuation with existing DMD-based electrostatic actuators. We would like to clarify that the micromirror demonstrated in our work operates on a fundamentally different principle compared to DMD devices. DMD-based systems rely on electrostatic forces to tilt micromirrors at short distances, primarily for light polarization and modulation. In contrast, our micromirror achieves two-dimensional positional shifts, driven by far-field optical forces, enabling displacements of up to tens of micrometers. This range of motion is significantly larger than that of traditional electrostatically driven devices, which are constrained to short-distance operation. Therefore, while DMD-based actuators excel in precision and reliability for specific applications, our approach offers unique advantages in scenarios requiring larger displacements and operation in dynamic or fluidic environments.

We acknowledge that further optimization is needed to match the durability and precision of DMD systems in dry environments, but our method expands the possibilities for applications where traditional electrostatic actuators are less suitable. We have revised the manuscript to include this comparative discussion, highlighting the distinct capabilities and potential applications of our system.

To clarify this, we have revised the manuscript to include this discussion:

Our system enables two-dimensional mirror shifts via light, achieving displacements up to tens of micrometers. It can serve as an alternative to established electrostatic actuator micromirrors for applications requiring larger displacement, expanding possibilities for dynamic or fluidic applications.

- 6. *The manuscript acknowledges localized heating at high light intensities, leading to nonlinear behavior. A more thorough analysis is needed, including:*
 - *Quantitative temperature rise estimates for different light intensities.*
 - *Potential long-term thermal degradation effects.*
 - *Strategies to mitigate heating in biological applications.*

Our response: We thank the Referee for these valuable suggestions.

Quantitative temperature rise estimates for different light intensities

Regarding the quantitative estimates of temperature rise for different light intensities, we have performed a COMSOL simulation to obtain the temperature at varying intensities. In the simulation, the temperature profile around the micromotor is evaluated by modeling the a-Si metaatoms as the primary sources of photothermal heating, incorporating material properties determined from ellipsometry. The heat source density is derived from absorption cross-section simulations, and the model accounts for both heat conduction and convection with appropriate boundary conditions. More details about the simulation can be found in the Methods section “Photothermal Heating Simulation”. The calculated temperature as a function of light intensities in four motors with varying amounts of meta-atoms is presented in Supplementary Figures 4e-h. The corresponding viscosity of water at different temperatures has also been calculated.

FIG. S4: **Finite element simulation of temperature distribution in four motors with varying amounts of meta-atoms.** **a-d** Absolute temperature distribution, obtained from finite element simulation, around a motor with **a** 55, **b** 36, **c** 29, and **d** 22 meta-atoms in each metasurface section illuminated with a $120 \mu\text{W } \mu\text{m}^{-2}$ s-polarized incident plane wave ($\lambda = 1064 \text{ nm}$). **e-h** Simulated temperature around a motor with **e** 55, **f** 36, **g** 29, **h** 22 meta-atoms, alongside calculated dynamic viscosity of water, as a function of incident light intensity.

Potential long-term thermal degradation effects

Regarding the long-term thermal degradation effects, the composite materials of the motors are Si, SiO₂ and SU-8 resist. As we can see from the Supplementary Figure 4, the highest temperature induced by the laser reported in our work is $\sim 40^{\circ}\text{C}$. At this temperature and in DI water, both Si and SiO₂ maintain their physical and chemical stability over long periods, without significant degradation or structural changes. The SU-8 photoresist has a glass transition temperature of approximately $\sim 200^{\circ}\text{C}$, and exposure to 40°C does not have a significant impact on its properties. Long-term immersion in water can result in slight swelling due to water absorption; however, the resulting surface roughness is negligible in our system. As demonstrated in [*Japanese Journal of Applied Physics*, 2014, 53, 08MC03] where the swelling was measured using atomic force microscopy (AFM), under conditions of 95% relative humidity, 80°C , and 500 hours of exposure, the observed swelling was less than 4 nm, which does not compromise the adhesion of the SU-8 photoresist to the SiO₂ substrate. Therefore, the system we developed is highly stable against potential long-term temperature-induced degradation effects.

To clarify this, we have revised the manuscript to mention this:

Regarding motor durability, although the chip is currently not optimally packaged to ensure long-term stability of the sample, the motor remains stable and operates effectively under illumination for eleven hours (Supplementary Video 4), with no observed degradation during this period.

Strategies to mitigate heating in biological applications

The system utilizes a 1064 nm laser, which is considered biologically safer due to its lower absorption by water and biological tissues compared to shorter wavelengths. This minimizes the potential for damage, making it suitable for biomedical applications.

The reported light intensity of $35\ \mu\text{W}\ \mu\text{m}^{-2}$ is based on a spot size of approximately $300\ \mu\text{m}$. However, since the driving gear, which powers the entire system, has a diameter of only $16\ \mu\text{m}$, the light can be precisely focused on this smaller area. Consequently, the total power required is only about 7 mW, a relatively low value that remains well within safe limits for biomedical use.

Furthermore, the light is directed solely to the driving gear, which subsequently transfers motion to passive structures. These passive structures, rather than the light

itself, interact with the biological samples. This design ensures that biological samples are not directly exposed to the laser, thereby mitigating concerns regarding potential light-induced damage.

Based on these considerations, we have clarified this aspect in the discussion of our manuscript.

By using light as a widely available and biocompatible energy source, these micromotors are well-suited for manipulating biological matter, including bacteria and cells. The system employs a 1064 nm laser, which minimizes damage to biological samples due to its low absorption by water and tissues. The light can be focused from a large area onto the small driving gear, operating at a low power requirement of just a few mW, which remains well within the safe thresholds for biological systems. Importantly, the light can be selectively directed to the driving gear, allowing it to mechanically actuate passive structures without directly exposing biological samples to the light source. This non-toxic, indirect energy delivery mechanism broadens the applications of our light-driven micromotors and metamachines in biomedical environments.

7. For readers familiar with optics, nanofabrication, or micromechanics, the paper is technically well-structured. However, for an interdisciplinary audience, some key concepts lack introductory explanations. For example, the paper does not provide an intuitive introduction to how optical metasurfaces convert light into motion. A short conceptual explanation of light-momentum transfer would improve readability.

Our response: We thank the Referee for this suggestion. We have revised the manuscript to include a discussion on momentum transfer by the optical metasurface to induce microobject motion.

Recent advances in active matter have used unfocused light to propel microscopic vehicles employing plasmonic or dielectric metasurfaces that generate lateral optical forces through directional light scattering. For example, it has been shown that microvehicles with plasmonic or dielectric nanostructures arranged in a parallel pattern can move forward under linearly polarized light via linear momentum transfer (*Science Advance*, 2020, 6, eabc3726; *Nature Nanotechnology*, 2021, 16, 970-971) and can be steered using polarized light through spin angular momentum transfer (*Nature Nanotechnology*, 2021, 16, 970-971). Furthermore, it has also been shown that arranging

the scatterers in a circular pattern enables rotation under linearly polarized light (*Science Advance*, 2020, 6, eabc3726; *Light: Science & Applications*, 2025, 14, 38). More advanced designs incorporate four individually addressable chiral plasmonic nanoantennas, allowing full 2D motion control through the application of dual-wavelength light. (*Nature Nanotechnology*, 2022, 17, 477-484).

- 8. *The study employs a specific gear design, but have the authors considered alternative tooth profiles? For example, would using involute or cycloidal profiles, commonly used in macroscopic mechanical gears, reduce inter-gear friction and improve transmission smoothness? Could using different materials for the metasurface enhance efficiency or reduce power requirements? A discussion of alternative design choices and their trade-offs would broaden the manuscript's applicability.*

Our response: We thank the Referee for highlighting that involute or cycloidal tooth profiles would be a superior design choice compared to the rectangular teeth used in this study, as they provide more reliable torque transfer and reduced friction. We will incorporate this design in future studies and have added the following paragraph to the discussion section, where we outline potential design optimizations for future work.

Beyond this, incorporating involute or cycloidal tooth profiles, commonly used in macroscopic gears, could reduce inter-gear friction and improve transmission efficiency. Alternative metasurface materials, such as TiO_2 , could extend the operational wavelength into the visible light region, simplifying optical calibration, while optimizing the metasurface design into grating or three-dimensional structures could enhance diffraction efficiency. These advancements would collectively improve the system's performance, adaptability, and applicability across diverse environments.

Response to the Reviewers' reports

We thank the Reviewers for their constructive comments and suggestions. We have carefully addressed each of their points in our response, making the necessary revisions to strengthen the manuscript.

Below, we provide a detailed point-by-point response to the Reviewers' comments, including references to the specific pages in the revised manuscript where the changes have been incorporated.

RESPONSE TO REVIEWER #1

- *The authors provided additional information to improve the manuscript and addressed part of my concerns in this revision.*

Our response: We thank the Reviewer for the constructive comments and for acknowledging the improvements made in this revision. We have addressed the remaining comments as explained below.

- *Still, I have some comments listed below:*

1. As shown in Supplementary video 4, the geometry of single metarotor almost remains unchanged while the rotation speed decreases after long-term illumination. It seems that the rotor is not as stable as the authors claimed. More discussions are expected here and the rotor performance should be objectively described in the revised manuscript. In addition, SEM images of the rotor at initial stage and after continuous irradiation are preferred to better clarify the degradation or not.

Our response: The Reviewer correctly points out that the rotation speed of our motors decreased over the course of an 11-hour continuous illumination experiment before eventually coming to a halt (Supplementary Video 4). Following the Reviewer's suggestion, we have now carefully investigated the factors responsible for the slow down of the motor during prolonged operation.

As suggested by the Reviewer, we acquired SEM images of the motor at two time points: (1) at the initial stage and (2) after 10 hours of prolonged operation and

irradiation. These images have been included in Supplementary Figure 9, shown below. As can be seen in the figure, we observed no visible structural changes or signs of degradation between the two time points, indicating that the motor remains intact.

Since the motor did not show any obvious damage from the SEM images, we hypothesized that the gradual slow down was due to gradual changes in the surrounding solution environment rather than to degradation or instability of the motor itself. In fact, light-induced thermal effects or a slow redistribution of surfactant molecules over time may alter the interfacial conditions. (As radiation pressure continuously presses the motor against the substrate, we rely on surfactants, such as Triton X-100, to reduce friction and enable sustained rotation. If the effectiveness of this lubrication layer is compromised, for example through local depletion or reorganization of the surfactant, the motor may experience increased interaction with the substrate, thereby hindering its rotation.)

To verify this hypothesis, we conducted the following additional experiment: After eleven hours of operation during which the motors slowed down, we carefully rinsed the chamber containing the motors using deionized water, added some new solution of water with 0.005 wt% Triton X-100, sealed the chamber, and sonicated it in an ultrasound bath. Upon illumination, the motor resumed rotation as at the beginning of the experiments. This observation, in combination with the SEM images, indicates that the decline in performance is reversible and likely caused by changes in the solution environment rather than by structural damage to the rotor.

We acknowledge that elucidating the mechanism by which changes in the solution environment affect the rotational behavior of the motor is of significant scientific interest. However, a comprehensive investigation (such as quantifying interfacial forces, characterizing nanoscale variations in motor–substrate separation, or understanding the interplay between surfactants, the motor, and the substrate) is beyond the scope of the present study. We intend to explore these aspects in some future work.

To address this point, we made the following changes to the manuscript: We added Supplementary Figure 9, with SEM images comparing a motor before and after illumination. In addition, we extended the Supplementary Movie 4 to include the renewed rotation after a cleaning step.

FIG. S9: **Motor durability.** **a** SEM image of a single motor before laser irradiation. **b** SEM image of motor arrays after laser irradiation for eleven hours and natural aging over six months. The right panel shows a zoom-in view of a single motor. The motor structure remained undamaged when compared to that of the motor shown in panel **a**.

We made the following changes to the manuscript:

Regarding motor durability, although the chip is not yet optimally packaged for long-term stability, the motor remains operational under continuous illumination for up to eleven hours (Supplementary Video 4). Furthermore, it does not undergo structural degradation even when irradiated for eleven hours and stored for up to six months (Supplementary Fig.S9). Nevertheless, while the motor is in operation, its rotational speed gradually decreases and eventually the motor stops. This is likely due to changes in the solution environment (such as local surfactant redistribution and the accumulation of impurities) leading to increased friction at the motor–substrate interface. Notably, the motor can resume rotation after gentle cleaning and solution exchange, indicating that these effects are reversible and can be mitigated through improved packaging and fluid handling.

- *2.Regarding maximum output torque or durability, the authors claim that light intensity can continuously increase the torque without a clear upper limit. However, high laser power may attribute to increased localized temperature, thermal fluidics or even bubbles to deteriorate the rotor performance, thus leading to an upper limitation of the maximum torque or durability.*

Our response: We thank the Reviewer for pointing this out. At higher light intensities, the optical torque naturally increases in a linear manner, as described by the

relationship:

$$\tau = 4r_0 \frac{n}{c_0} P_0 (T^{+1} - T^{-1} + R^{+1} - R^{-1}) \sin(\theta)$$

where T^{+1} and T^{-1} denote the transmission efficiencies of the +1 and -1 diffraction orders, respectively; R^{+1} and R^{-1} denote the reflection efficiencies of the corresponding orders; θ is the diffraction angle; r_0 is the moment arm; n is the refractive index; P_0 is the light power; and c_0 is the speed of light in vacuum. (See Supplementary Note: “Rotation Mechanism of the Motor” for further details.)

However, as the Reviewer rightly noted, excessively high light intensity can cause significant heating, induce convection and bubble formation, and potentially damage the motor. These effects can limit the torque and durability achievable in practical operation. We apologize for not addressing this in the original manuscript, which may have led to a misunderstanding. To clarify, we have revised the text and now reads:

Under the intensity of $88.5 \mu\text{W} \mu\text{m}^{-2}$, we measure a maximum torque of $36 \text{ pN} \cdot \mu\text{m}$ obtained from $\tau = \omega\gamma_r$. It is important to note that there is an upper limit to the applicable torque, as higher laser powers can induce local heating and bubble formation (see Supplementary Note “Rotation mechanism of the motor” for details).

- 3. Page 16, “in this word, tens of thousands of micromotors were fabricated on a $5 \text{ mm} \times 5 \text{ mm}$ area”. SEM images of the rotor array with thousands of rotors are required to show the scalability.

Our response:

Due to the limited field of view of a high-resolution scanning electron microscope (SEM), it is technically challenging to capture thousands of micromotors in a single image. To effectively demonstrate scalability, we provide an optical image of the entire chip ($15 \text{ mm} \times 15 \text{ mm}$), which is used for follow-up studies. A $5 \text{ mm} \times 5 \text{ mm}$ region within this chip already contains tens of thousands of micromotors arranged in regular arrays, thus demonstrating the platform’s scalability.

Using optical microscopy, we imaged a localized area ($1.33 \text{ mm} \times 1.07 \text{ mm}$) of the motor array, capturing 1,280 micromotors in a single frame. Figure S2b shows the largest field of view obtainable with our Scanning Electron Microscope (SEM - Zeiss Supra

FIG. S2: **Thousands of light-driven micromotors fabricated in parallel.** **a** Optical image of a $15\text{ mm} \times 15\text{ mm}$ glass chip containing $5\text{ mm} \times 5\text{ mm}$ motor array areas fabricated in parallel. The total number of motors is 22,500. The right panel shows a magnified microscope image highlighting 1,280 individual motors. **b** SEM image of a large motor array containing 952 motors. The right side shows zoomed-in SEM images of a portion of this array.

60 VP), capturing 952 micromotors at once. This image has been magnified to reveal detailed structural information.

Regarding the manufacturing process, the most time-consuming step is the electron beam lithography used to write the metaatoms. Our equipment (Raith EBPG 5200) is capable of patterning a $5\text{ mm} \times 5\text{ mm}$ mm area in approximately 10 minutes.

We have revised the manuscript to include this result as Supplementary Figure 2. As shown above and modified the main text:

Using this approach, tens of thousands of micromotors can be fabricated within a $5\text{ mm} \times 5\text{ mm}$ area on a single chip. Moreover, the fabrication process can be scaled up to the wafer level. A comprehensive explanation is provided in the Methods section and Supplementary Fig. ??, S2.

- 4. In this revision, the authors seem to avoid the discussion on power efficiency or energy conversion efficiency of the rotor, which is a key criterion for judging its performance. More in-depth and quantitative analysis are needed.

Our response: We thank the Reviewer for pointing this out. The energy conversion efficiency (ϵ) of the motor is given by $\epsilon = \frac{P_{\text{out}}}{P_{\text{in}}}$. The input power is calculated as $P_{\text{in}} = IA$, where I is the light intensity and A is the area of the metasurface. The output power is given by $P_{\text{out}} = \tau\omega$, where τ and ω are the torque and angular velocity, respectively. The torque is calculated using $\tau = \omega\gamma_r$, where ω is obtained from the experimental data, and γ_r is the rotation friction, which can also be calculated. (see Supplementary Note: “Rotation Mechanism of the Motor”).

We have now added this calculation in the Supplementary Material:

The energy conversion efficiency (ϵ) of the motor can be calculated as:

$$\epsilon = \frac{P_{\text{out}}}{P_{\text{in}}},$$

The output power is given by:

$$P_{\text{in}} = IA$$

where I is the light intensity and A is the area of the metasurface. The output power is given by:

$$P_{\text{out}} = \tau\omega = \gamma_r\omega^2$$

where τ and ω are the torque and angular velocity, respectively. Using experimental values, for example, under illumination with an intensity of $88.5 \mu\text{W} \mu\text{m}^{-2}$, the input power is $P_{\text{in}} = 16 \text{ mW}$, the output power is $P_{\text{out}} = 4.8 \times 10^{-13} \text{ mW}$, and the efficiency is $\epsilon = 3.0 \times 10^{-14}$.

Based on these expressions, we can estimate the energy conversion efficiency of the motor to be on the order of $\sim 10^{-14}$.

We compared the energy conversion efficiency with other micromotor driving methods and included it as a column (ϵ) in Table 1 (note that for many micromotors in the literature this , as shown below:

Table S1: Comparison of mechanisms to drive micromotors

Driving methods	Material of motors	Medium	Dimension of individual motors	Reported maximum rotation speed	Addressable control	Coupling motors to functional microscopic geared mechanisms		ϵ	Ref
						Size	Coupling methods		
Static	Si, SiO ₂ , Si ₃ N ₄	Air	120 μ m diameter	500 rpm	NA (Difficult)		No	NA	[1]
Static	Carbon Nanotubes	Air	2 μ m in length	NA	NA (Difficult)		No	NA	[2]
Static	SU8	oil	100 μ m diameter	3000 rpm	NA (Difficult)		No	NA	[3]
Static (Comb drive)	Si, SiO ₂	Air	50 μ m diameter	\sim 200000 rpm	NA (Difficult)	> 100 μ m	Completely batch-fabricated	NA	[4]
Ac electric	Au@Ni nanorods	DI water	10 μ m in length	18000 rpm	NA (Difficult)		No	NA	[5]
Ac electric	Au@SU8	DI water	\sim 25 μ m diameter	\sim 23 rpm	NA (Difficult)		No	NA	[6]
Magnetic	Fe ₂ O ₃ particles @resist	DI water	\sim 30 μ m diameter	\sim 150 rpm	NA (Difficult)		No	NA	[7]
Magnetic	NdFeB particles @resist	DI water	\sim 100 μ m diameter	\sim 720 rpm	NA (Difficult)	\sim 300 μ m	Completely batch-fabricated	NA	[8]
Chemical	Pt@resist	H ₂ O ₂ /H ₂ SO ₄ /DI water	120 μ m diameter	60 rpm	NA (difficult)		No	NA	[9]
Chemical	Pt particles@resist	H ₂ O ₂	\sim 30 μ m diameter	\sim 160 rpm	NA (Difficult)	\sim 50 μ m	Completely batch-fabricated	NA	[10]
Micro-organisms	SU8	Buffer	\sim 15 μ m diameter	\sim 15 rpm	Yes		No	$\sim 10^{-5}$	[11]
Optical (OET)	SU8	DI water	150 μ m diameter	\sim 83 rpm	Yes	> 300 μ m	Piece-part assembly	NA	[12]
Optical	Polymer (3D-shaped)	DI water	4.5 μ m diameter	NA	Yes	\sim 15 μ m	Piece-part assembly	$\sim 10^{-13}$	[13]
Optical (nanoantenna)	Au@resist	DI water	2.5 μ m diameter	\sim 190 rpm	Yes		No	$\sim 10^{-16}$	[14]
Optical (metasurface)	Si, SiO ₂	DI water	8 μ m diameter	\sim 240 rpm	Yes		No	$\sim 10^{-14}$	[15]
Optical (metasurface)	Si, SiO ₂ , SU8	DI water	\sim 10 μ m diameter	120 rpm	Yes	\leq 50 μ m	Completely batch-fabricated	$\sim 10^{-14}$	This work

Table notes: 1. Static (Combe drive): Microelectromechanical Systems (MEMS) linear actuators using electrostatic forces between conductive combs. 2. Optical (OET): OptoElectronic Tweezer. 3. The table column “Coupling motors to functional microscopic geared mechanisms” is split into “Size” and “Coupling methods.” “Size” indicates the overall dimensions of the microscopic geared mechanisms, while “Coupling methods” specifies how motors are connected to them. 4. ϵ : Energy conversion efficiency.

Although this value is very low compared to some biological motors, such as those found in bacteria (*Ref [11]: Nat Commun, 2021,12, 5349*), this is expected because our motor is driven by the momentum of photons rather than by heat, chemical reactions, or other mechanisms. The efficiency is consistent with values reported for systems driven by optical angular momentum (*Ref [14]: Nature Nanotechnology, 2022, 17, 477; Ref [15]: Light: Science and Applications, 2025, 14, 38*). This is due to the weak intrinsic momentum of photons ($p = \frac{h}{\lambda} \sim 10^{-28} \text{ kg} \cdot \text{m/s}$), as well as inevitable energy losses due to scattering and viscous drag, which fundamentally limit conversion efficiency. While chemically powered motors benefit from the dense energy release of molecular reactions, optical driving offers unique advantages such as contactless control and precision manipulation, making this trade-off acceptable for targeted applications. As emphasized in the article, our approach focuses on constructing a stable mechanical system consisting of multiple coupled units. This is difficult to achieve using other methods (see Supplementary Material, Table 1). Regarding this mechanical energy conversion, the system of two gears with different sizes shown in Figure 2, for example, can reach very high energy conversion efficiency.

To clarify this, we have revised the manuscript to add this:

The energy conversion efficiency (ϵ) of the motor can then be calculated as $\epsilon = P_{\text{out}}/P_{\text{in}} = \tau\omega/P_{\text{in}}$. The order of magnitude of the conversion efficiency is approximately 10^{-14} , which is consistent with previously reported motors driven by light momentum [*Nature Nanotechnology, 2022, 17, 477; Light: Science and Applications, 2025, 14, 38*]. A comparison of efficiencies is provided in Table S1.

and also in Supplementary material:

Recent advancements have demonstrated motors driven by nanophotonic structures

(*Nature Nanotechnology*, 2022, 17, 477; *Light: Science and Applications*, 2025, 14, 38), which operate primarily through the transfer of momentum between incident light and the structure to induce rotational motion. Although the energy conversion efficiency of such optically driven motors remains lower than that of conventional electric or magnetic field driven counterparts, their capabilities (fully batch-fabricated production, addressable control, and scalable manipulation) offer compelling solutions to persistent challenges in the scalable manufacturing, precise control, and integration of microscale and nanoscale motors. However, existing research has largely focused on constructing freely rotating motors in solution, rather than on functional micro- and nanoscale machines. This work overcomes these limitations, successfully achieving scalable manufacturing and integration of micro- and nanoscale machines.

- 5.Picky, “operates effectivelu” should be “operates effectively”.

Our response: We thank the Reviewer for pointing this out. We have corrected this typo in the manuscript.

RESPONSE TO REVIEWER #2

- *The authors have made extensive changes and properly addressed all my concerns. The only minor suggestion I have now is to provide a reference to the claim that their light power is "well within the safe thresholds for biological systems".*

Our response: We thank the Reviewer for the constructive comments and for acknowledging the improvements made in this revision. We have revised the manuscript to add a reference (*Rotational optical tweezers for active microrheometry within living cells. Optica, 2022, 9, 1066-1072.*). In this work, they used a laser power much higher than ours to operate in living cells, while still maintaining cell activity. And in our earlier work (*Microscopic metavehicles powered and steered by embedded optical metasurfaces. Nature Nanotechnology, 2021, 16, 970-974.*), we used the same design of particles to propel yeast cells. These support the claim that the light power used is well within the safe thresholds for biological systems.

To clarify this, we have added the references to the manuscript, which now read as:

The system employs a 1064 nm laser, which minimizes damage to biological samples due to its low absorption by water and tissues [*Optica, 2022, 9, 1066-1072; Nature Nanotechnology, 2021, 16, 970-974.*].

This manuscript builds upon prior work (Nature Nanotechnology 16, 970–974 (2021)) by further extending the application of programmable optical metasurfaces to microscopic gear mechanisms. Through several micro-mechanical models, the study effectively demonstrates the gear system's high motion performance, precise controllability, and scalability, creating a practical and multifunctional platform for micro- and nanoscale mechanical systems.

Overall, the work presents an innovative approach to microscale gear mechanisms with strong potential. However, I recommend major revisions to address my concerns listed below to improve the manuscript's impact, clarity, and broader relevance.

1. Practical Limitations

- **High Light Intensity Requirement**
The system requires a light intensity of approximately $35 \mu\text{W}/\mu\text{m}^2$ to achieve typical rotational speeds. This necessitates complex external optical setups, which could limit real-world applications. A discussion on potential power reduction strategies or trade-offs would be valuable. In addition, is this power level safe for biomedical applications?
- **Limited Motion Adjustability**
The reliance on pre-designed optical metasurfaces somewhat restricts the flexibility in dynamically modifying gear motion modes. The manuscript should acknowledge this limitation and explore possible workarounds, such as reconfigurable metasurfaces or adaptive optics.
- **Scalability and Large-Scale Fabrication**
While the manuscript highlights CMOS compatibility, it lacks quantitative discussion on scalability. Specifically:
 - How many micromotors can be integrated on a single chip?
 - Are there fabrication constraints that could hinder large-scale production?
 - How does optical alignment complexity scale as the number of micromotors increases?

2. Comparison with Competing Technologies

The manuscript introduces metasurface-powered gears as a superior alternative to existing methods, but the justification remains qualitative. A more quantitative comparison with other techniques is necessary.

An additional class of work has explored electrostatic-driven micromechanical gears (e.g., <https://doi.org/10.3311/PPch.10274>). These do not require complex electrical connections around the gears, enable large-scale parallelization, can be fabricated using standard photolithography, and do not require intricate metasurface designs or external optical control. I encourage the authors to consider: How does the power efficiency of metasurface-driven gears compare to electrostatic gears? Can the scalability and manufacturability of these systems be directly compared?

Other studies, such as Catchmark, Jeffrey M.; Subramanian, Shyamala; Sen, Ayusman, Directed rotational motion of microscale objects using interfacial tension gradients continually generated via catalytic reactions, *Small* (2005), 1 (2), 202-206,

have demonstrated chemically powered microgears. These achieve similar functionalities and do not require external optical control.

Finally, the manuscript claims that optoelectronic tweezers (OET) “lack flexibility,” making them inferior to metasurface-driven systems. However, OET demonstrations show similar functionalities to those presented here. Could the authors provide a head-to-head, quantitative comparison with OET? Where does the proposed technique outperform OET, and in which aspects might OET remain more practical?

A comparative table, and a dedicated discussion, summarizing advantages and trade-offs across these technologies would significantly improve the manuscript’s clarity and scientific positioning. I do not expect the current technique to win it all, but it would be very useful to see where it wins and where other technologies might be more useful.

3. Theoretical Predictions of Force and Torque

The manuscript lacks a theoretical model or numerical simulation for the expected force and torque generated by the metasurface-driven micromotors. Instead, it relies on experimental measurements and qualitative analysis. I wonder if the authors can derive force and torque estimates based on optical momentum transfer and fluid dynamics? Would finite element modeling (FEM) or a simplified analytical model help predict performance under different conditions?

Additional Recommendations

1. The manuscript states that increasing the number of meta-atoms raises the local temperature, reducing viscous drag. This should, in principle, lead to a superlinear increase in rotational speed, rather than the observed sublinear trend. However, the manuscript attributes the slowdown to localized heating effects. Can the authors clarify what specific factors (e.g., light absorption saturation, thermal expansion, changes in optical torque efficiency) contribute to this behavior, and whether they have been quantitatively modeled?
2. Additional schematic illustrations marking the gap position and probability distribution on Fig. 1f/g would improve reader comprehension.
3. In Supplementary video 11, indicating the number of metasurfaces incorporated into each of the three racks would enhance clarity.
4. While the manuscript discusses metasurface quantity and light intensity effects on speed, it does not address maximum output torque or durability. Referencing Refs. 24 and 25, which analyze similar micromechanical systems, would strengthen the discussion.
5. The authors suggest applying this gear mechanism to mirror actuation. How does its durability and precision compare to existing DMD-based electrostatic actuators?
6. The manuscript acknowledges localized heating at high light intensities, leading to nonlinear behavior. A more thorough analysis is needed, including:
 - Quantitative temperature rise estimates for different light intensities.
 - Potential long-term thermal degradation effects.
 - Strategies to mitigate heating in biological applications.

7. For readers familiar with optics, nanofabrication, or micromechanics, the paper is technically well-structured. However, for an interdisciplinary audience, some key concepts lack introductory explanations. For example, the paper does not provide an intuitive introduction to how optical metasurfaces convert light into motion. A short conceptual explanation of light-momentum transfer would improve readability.

8. The study employs a specific gear design, but have the authors considered alternative tooth profiles? For example, would using involute or cycloidal profiles, commonly used in macroscopic mechanical gears, reduce inter-gear friction and improve transmission smoothness? Could using different materials for the metasurface enhance efficiency or reduce power requirements? A discussion of alternative design choices and their trade-offs would broaden the manuscript's applicability.